# Value-aligned Behavior Cloning for Offline Reinforcement Learning via Bi-level Optimization

**Xingyu Jiang**[1]**, Ning Gao**[1]**, Xiuhui Zhang**[1]**, Hongkun Dou**[1]**, Yue Deng**[1,2] *

[1] Beihang University, [2] Beijing Zhongguancun Academy

`{jxy33zrhd,gaoning_ai,zhangxiuhui,douhk,ydeng}@buaa.edu.cn`

## Abstract

Offline reinforcement learning (RL) aims to optimize policies under pre-collected data, without requiring any further interactions with the environment. Derived from imitation learning, Behavior cloning (BC) is extensively utilized in offline RL for its simplicity and effectiveness. Although BC inherently avoids out-of-distribution deviations, it lacks the ability to discern between high and low-quality data, potentially leading to sub-optimal performance when facing with poor-quality data. Current offline RL algorithms attempt to enhance BC by incorporating value estimation, yet often struggle to effectively balance these two critical components, specifically the alignment between the behavior policy and the pre-trained value estimations under in-sample offline data. To address this challenge, we propose the Value-aligned Behavior Cloning via Bi-level Optimization (VACO), a novel bi-level framework that seamlessly integrates an inner loop for weighted supervised behavior cloning (BC) with an outer loop dedicated to value alignment. In this framework, the inner loop employs a meta-scoring network to evaluate and appropriately weight each training sample, while the outer loop maximizes value estimation for alignment with controlled noise to facilitate limited exploration. This bi-level structure allows VACO to identify the optimal weighted BC policy, ultimately maximizing the expected estimated return conditioned on the learned value function. We conduct a comprehensive evaluation of VACO across a variety of continuous control benchmarks in offline RL, where it consistently achieves superior performance compared to existing state-of-the-art methods.

## 1 Introduction

Over the past decade, reinforcement learning (RL) has made remarkable advancements and exerted significant influence in domains such as robotics control (33), dynamic programming (49), and recommendation systems (6). However, in real-world scenarios where interactions are either extremely challenging or prohibitively costly, traditional online RL faces considerable limitations and may often prove impractical. As a result, offline RL has emerged as a promising alternative, attracting growing attention and active research. Unlike online RL, which actively collects trajectory samples through direct interaction with the environment, offline RL is restricted to deriving policies from a static, pre-collected dataset (30), without any further interactions with the environment. Interestingly, the primary advantage of offline RL, its independence from environment interactions, also poses its most significant challenge. Due to the limited and potentially sub-optimal nature of offline data samples, offline RL confront two severe challenges: 1) out-of-distribution (OOD) issues and 2) value alignment issues. Both of these issues fundamentally stem from inadequate sampling of the state and action spaces in offline datasets.

As illustrated in Fig.1, we provide a schematic diagram to intuitively depict the two critical challenges in offline reinforcement learning. Fig.1(a) presents the learned value-action curve for a given state, where the blue dots represent actions collected from the offline dataset, which we refer to as the *in-sample domain*. The adjacent red area, representing actions that have not been collected, is

---
*Corresponding author

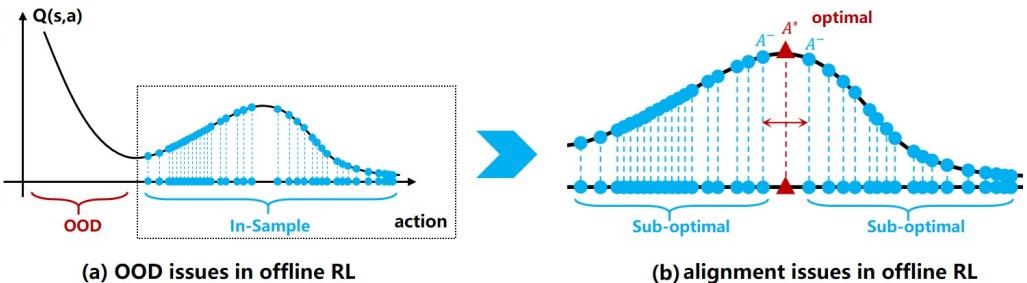

Figure 1: Illustration of the two challenges in offline RL: (a) value overestimation in OOD issues and (b) sub-optimal policy extraction in value alignment issues.

termed the *out-of-distribution (OOD) domain*. Empirically, the values of actions in the OOD domain tend to be overestimated, while the action values within the in-sample domain are more confident and closer to their true values. Fig.1(b) further refines the in-sample domain. During the behavior policy learning process, the inadequacy of action space often causes the behavior policy to converge on sub-optimal actions within the in-sample domain (e.g., $A^-$), which actually do not align with the optimal actions suggested by the value estimations (e.g., $A^*$). This misalignment can lead to sub-optimal policy performance.

To avoid above challenge, behavior cloning (BC (37)) is initially employed in offline RL settings. BC utilizes supervised learning to directly learn from the in-sample data, effectively avoiding OOD challenge. However, BC lacks the capability to discern between high-quality and low-quality data, which typically results in competitive performance only with expert offline datasets and leads to failure within poor/sub-optimal offline datasets. To circumvent this limitation, recent offline RL approaches have begun incorporating certain auxiliary information (e.g., value function) into BC, guiding it towards differentiated learning. These methods can be well categorized into following primary types: (1) **explicit regularization constraints** (13; 27), which involve heuristic regularization terms like KL divergence or expectile regression to directly constrain the behavior policy; (2) **implicit regularization constraints** (22; 55), which leverage generative models such as VAEs (9; 29) and diffusion models (19; 44) to enforce constraints on the behavior policy within latent action space; and (3) **return-conditioned supervised learning** (5; 23), which utilizes decision transformer to directly model dynamic programming process in a conditioned supervised learning manner. Although the aforementioned methods have achieved notable performance, they often struggle to balance the OOD challenge and value alignment issue concurrently.

**Our contribution.** In this paper, inspired by recent advancements in meta-learning (34; 41; 21; 52)), we introduce the Value-aligned Behavior Cloning via Bi-level Optimization (VACO), a novel bi-level offline RL framework to balance the OOD and alignment challenges. This framework integrates a simple multi-layer perception (meta-scoring network) to assign differential importance weights to various state-action sample pairs. In this way, the conventional BC loss function is transformed into a weighted summation of individual sample losses, enabling differentiated learning for varying quality samples. Such learning manner is intuitive and closely mirrors human behavior learning processes. In detail, the internal loop of VACO executes weighted supervised behavior cloning with the assigned scores of in-sample data; and the external loop of VACO maximizes value estimation for value alignment and introduces controlled noise to enable limited exploration. Such bi-level configuration allows VACO to identify the optimal weighted BC policy, ultimately maximizing the expected estimated return conditioned on the learned value function without numerous hyper-parameters or complicated secondary components, such as generative models and transformer. It is worth mentioning that the meta-scoring network utilized in VACO is operational solely during the training phase and can be deactivated during testing, ensuring that the inference speed of the algorithm remains unaffected.

We evaluated the proposed VACO on the D4RL benchmark dataset (11) for continuous control tasks. The results indicate that our model achieves state-of-the-art (SOTA) performance, surpassing the above three categories of offline reinforcement learning algorithms, through a simple integration of BC (37) with DPG (42) algorithm in bi-level manner. Such achievement provides new insights

on the potential of bi-level optimization in offline RL, and further ablation studies underscore the effectiveness of our proposed meta-scoring network.

## 2 RELATED WORK

In recent years, offline RL has emerged as a pivotal solution for real-world scenarios, particularly in domains such as autonomous driving (43), dynamic programming (49), and robotics control (33). Offline RL is inherently limited by insufficient data and the lack of interaction with the environment, leading to two major challenges: 1) the OOD problem and 2) the value alignment issue. Traditional offline methods (e.g., BC (37) and TD3 (14)) often focus on one specific aspect. More recent strategies (15; 54; 24; 47; 56) seek to integrate these approaches for performance boosting. Depending on the integration manner, these approaches can generally be classified into three categories:

**Explicit Regularization** directly incorporates BC as a regularization within value estimation. For instance, TD3+BC (13) directly combines the loss function of BC and TD3, with hyperparameters fine-tuning the balance between them. BRAC+ (53) integrates BC into the value and policy update processes by KL divergence (18) to provide regularization and constraint. IQL (27) employs expected regression to perform in-sample value estimation, subsequently guiding BC across various state-action pairs using advantage-weighted regression (AWR) (36). This category of methods typically designs a divergence measure to ensure that the learned policies closely with the dataset's sampling policy to some extent.

**Implicit Regularization** posits that constraints should be moderately stringent, often executed within a latent space to subtly constrain policies. Key contributions in this area include MOPO (51), which introduces model-based concept into offline settings by incorporating penalty rewards to limit the learning process; PLAS (55), which uses a CVAE (2) to model policies within the action latent space, thus implicitly restricting learning to the support range of the in-sample dataset; and EDP (22), which utilizes diffusion models for implicit action estimation. These methods (20) typically employ generative models to encode state and action into latent space, thereby constraining learned policies.

**Return-conditioned Supervised Learning** represents a novel paradigm in offline RL that conditions the policy not only on the current state but also on expected future returns. This can be viewed as a variant of conditional BC. For instance, DT (5) introduced the use of decision transformers and return-to-go to achieve supervised modeling of offline trajectories;DS4 (8) innovates by replacing the time-invariant state-space layers with transformer, thereby facilitating efficient dynamic modeling. DC (23) emphasizes the importance of local attention in Markov processes and implements an innovative approach by substituting attention mechanisms with convolution.

**Bi-level Optimization in Offline RL.** Bi-level optimization is committed to optimizing another set of parameters other than target network parameters, which describes higher-level elements related to training neural networks(52; 21). In offline RL domain, the most relevant work to us is (56) and we differ in motivation and upper-lower functions as follows: (1)Motivation: we introduce the bi-level optimization framework, aiming to balance the OOD and value alignment problems,while (56) mainly focuses on the distributional shift issue. (2)Upper-lower functions: Different motivations lead to different upper and lower functions. We first introduce a meta-scoring network in the upper function to assign adaptive weights and mainly focus on the behavior policy updating in the lower function. Differently, (56) adopts the behavior policy updating in the upper function and mainly focuses on the Q value approximation in the lower function.

In our work, we explore the integration of BC and value estimation through a bi-level optimization framework, introducing a novel training methodology for offline RL to balance OOD and value alignment challenges concurrently. Although our proposed framework, VACO, is grounded in BC and DPG value estimation loss, it should better be regarded as a flexible framework capable of bridging supervised behavior policies with value estimation loss. With tailored modifications, this framework can be properly applied to the aforementioned three kinds of methods.

## 3 PRELIMINARIES

**Behavior cloning.** BC is an approach within imitation learning that trains policies to emulate expert behaviors by directly mapping observed states to corresponding actions. Typically, BC (37)

employs supervised learning models to approximate the policy demonstrated by the expert, effectively replicating the expert's decision-making process. Recently, behavioral cloning has gained popularity in offline RL due to its simplicity and straightforward application; however, its efficacy is heavily dependent on the quality and comprehensiveness of the demonstration data.

**Offline reinforcement learning setting.** In RL setting, the dynamic system is described in terms of a Markov decision process (MDP) setting, which is represented as a tuple $\mathcal{M} = \{\mathcal{S}, \mathcal{A}, r, \mathcal{P}, \rho, \gamma\}$, where $\mathcal{S}$ and $\mathcal{A}$ are the state and action space, $r(s, a)$ is a scalar reward function, $\mathcal{P}$ is the transition dynamics, $\rho$ is the initial state distribution, and $\gamma \in (0, 1)$ is a discount factor. The objective of RL is to learn a policy $\pi(a|s)$ with parameters $\phi$ by maximizing the expected cumulative discounted return $\mathbb{E}_\pi[\sum_{t=0}^\infty \gamma^t r(s_t, a_t)]$, which is typically approximated by a value function $Q(s, a)$ with parameters $\theta$. For actor-critic based methods (26; 17) in a continuous action space, the parameters $\theta$ is typically updated by minimizing the squared Bellman error with an experience replay dataset $\mathcal{D}$ and a target function:

$$J_Q(\theta) = \mathbb{E}_{(s,a,r,s')\sim D}[Q_\theta(s, a) - r - \gamma Q_{\bar{\theta}}(s', \pi_\phi(s'))]^2 \tag{1}$$

where $Q_{\bar{\theta}}$ denotes a target Q-function, which is a delayed copy of the current Q-function $Q_\theta$. Then, the policy $\pi_\phi$ can be updated following the deterministic policy gradient (DPG (42)) theorem:

$$J_\pi(\phi) = \mathbb{E}_{s\sim D}[-Q_\theta(s, \pi_\phi(s))] \tag{2}$$

In offline RL setting, the learning policy is constrained on a pre-collected dataset $\mathcal{D}(s, a, s', r)$ without further interaction with the environment during the learning process. Meanwhile, the dataset $\mathcal{D}$ is generated by a unknown behavior policy $\pi_\beta$. Directly applying standard RL methods in the offline setting suffers from severe OOD problem in value and alignment issue in policy. To avoid above challenges, a widely used offline RL framework (48; 7) adopt the following behavior regularization scheme which regularizes the divergence between the learned policy $\pi_\phi$ and the unknown behavior policy $\pi_\beta$ of the dataset $\mathcal{D}$:

$$\pi = \arg\min_\pi \mathbb{E}_{s\sim D}[-Q_\theta(s, \pi_\phi(s)) + D(\pi_\phi(\cdot|s)||\pi_\beta(\cdot|s))] \tag{3}$$

where $D(\cdot||\cdot)$ is some divergence measures, which can have either an explicit or implicit form, to constrain how *"closeness"* of the learned policy $\pi_\phi$ and the unknown behavior policy $\pi_\beta$.

## 4 METHOD: VALUE-ALIGNED BEHAVIOR CLONING WITH BI-LEVEL OPTIMIZATION

In this section, we first discuss the motivation for balancing OOD and alignment problem. Subsequently, we introduce how to transform the behavior cloning supervised loss function into a weighted behavior cloning supervised loss function. Finally, we describe how to correctly guide the proposed weighted supervised loss through bi-level optimization manner, thereby effectively balance the OOD issue and the alignment problem.

### 4.1 MOTIVATION FOR BALANCING OOD AND ALIGNMENT PROBLEM

In offline reinforcement learning (RL) settings, two classic approaches—behavior cloning (e.g., BC (37)) and value estimation (e.g., TD3 (14))—are commonly employed to address the OOD and value alignment challenges, respectively. The objective function of behavior cloning (see Eq.4) effectively mitigates OOD issues but suffers significantly from value misalignment. In contrast, the objective function of value estimation (see Eq.2) adeptly resolves alignment issues but is highly susceptible to OOD challenges. These phenomena are evident in our experimental results, as shown in Fig.2: behavior cloning consistently demonstrates stable performance across different datasets but tends to under-perform, while value estimation achieves superior results in specific datasets but displays extreme instability, sometimes resulting in *"zero"* performance in others. In offline RL, these dual challenges—OOD and value misalignment—significantly impact model performance, and it is crucial to strike an appropriate balance between them.

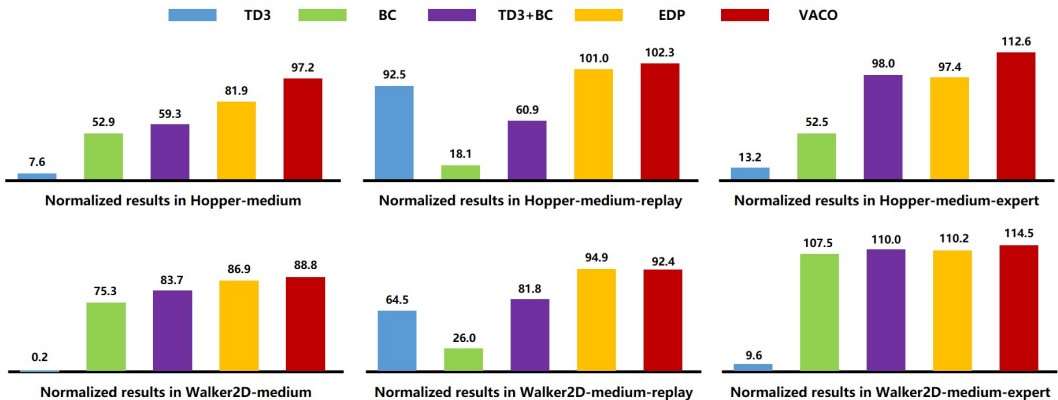

Figure 2: Normalized result comparisons on various Hopper and Walker2D datasets across TD3, BC, TD3+BC, EDP and VACO. Our method achieves superior performance.

Current offline RL algorithms (13; 53) attempt to balance the OOD issue and the alignment problem by using a mixed policy learning objective function, as represented in Eq.3, with notable success. However, the straightforward combination of behavior cloning (26) and deterministic policy gradient (DPG) (42) objective functions (e.g., TD3+BC (13)) can still lead to sub-optimal performance in some certain scenarios, as illustrated in Fig.2. To address this gap, we propose a novel bi-level optimization framework. In this framework, a weighted behavior cloning algorithm is employed in the inner loop for policy extraction, while a value estimation algorithm is adopted in the outer loop for value alignment. Additionally, we introduce a meta-scoring network to facilitate an indirect fusion between behavior cloning and value estimation. In the following sections, we will describe the bi-level framework in detail.

## 4.2 WEIGHTED BEHAVIOR CLONING

We start with behavior cloning. In conventional behavior cloning training, a batch of state-action pairs $(s, a)$ is sampled from the dataset $\mathcal{D}$ and fed to the policy network $\pi_\phi$. The policy network $\pi_\phi$ employs a supervised learning approach, utilizing simple L2 loss function to optimize the learning process. The objective function $J_{BC}$ can be expressed as follows:

$$J_{BC}(\phi) = \mathbb{E}_{(s,a)\sim D}[\pi_\phi(s) - a]^2 \qquad (4)$$

Above additive loss $J_{BC}$ implies that different state-action pairs are treated equally in training, although they are actually worse action choices. For a more reasonable loss design, the weights of different training pairs should be properly evaluated and exploited. Therefore, we introduce a more intuitive behavior cloning loss $J_{BC}^w$ by a meta-scoring network $w(s, a, Q_\theta(s, a))$ with parameters $\alpha$:

$$J_{BC}^w(\phi) = \mathbb{E}_{(s,a)\sim D}\{w_\alpha(s, a, Q_\theta(s, a)) \cdot [\pi_\phi(s) - a]^2\} \qquad (5)$$

We emphasize here that evaluating individual state-action pairs is extremely challenging, particularly because most sample trajectories in offline reinforcement learning are sub-optimal. This complexity arises from the inherent nature of the collected data, which does not always represent the optimal policy but rather a mixture of various policy executions, often leading to less than ideal decisions being captured in the dataset. Accordingly, rather than a heuristic manner, we opt to adopt the parameterized meta-scoring mechanism that can automatically assess the importance of each training pair through a learnable neural network $w_\alpha$ with $(s, a, Q_\theta(s, a))$ as input.

## 4.3 BI-LEVEL OPTIMIZATION FRAMEWORK

While the aforementioned parameterized weighting concept is simple, it yields a highly under-determined and non-convex objective function coupled with two unknown neural networks. Without

extra constraints, the direct minimization of $J_{BC}^w$ can easily lead to a trival solution. In this case, the meta-scoring neural network $w_\alpha$ may intend to assign (near) zero weights to all sample pairs and hence totally mute the functions of the policy network (see the multiplications between the weight term and behavior cloning loss term in Eq.5). To avoid such trivial solution, extra constraints or guiding information should be imposed to restrict the feasibility of the learned policy. In this work, aiming to balance the OOD problem and value alignment issue concurrently, we consider enforcing the feasibility of the learned policy by maximizing the value estimation function in Eq.2, thereby meanwhile achieving value alignment. With this objective, the whole learning process is subject to the following bi-level optimization with a controllable Gaussian noise $N(0, \sigma)$ for limited exploration:

$$\min_\alpha \quad J_\pi(\phi) := \mathbb{E}_{s \sim D}[-Q_\theta(s, \pi_\phi(s + N(0, \sigma)))]$$

$$\text{s.t. } \phi^*(\alpha) = \arg\min_\phi J_{BC}^w(\phi) := \mathbb{E}_{(s,a) \sim D}\{w_\alpha(s, a, Q_\theta(s, a)) \cdot [\pi_\phi(s) - a]^2\} \tag{6}$$

The above bi-level optimization is composed of the internal loop in the constraint and the external loop in the objective function. The internal loop minimizes the empirical squared error of the offline in-sample dataset under guidance of the different weights provided by the external loop. The external loop maximizes the value estimation of the learned policy to tune the parameter space of the meta-scoring network for better value alignment. Through such a bi-level optimization framework, we achieve policy extraction from weighted behavior cloning in the internal loop to avoid out-of-distribution challenges and maintain value alignment from value estimation maximization in the external loop, respectively.

In practice, we opt not to incorporate the learning of the value network $Q_\theta$ within our bi-level optimization framework, as detailed in Eq.6. Consistent with practices in IQL (27), in offline RL, the learning of the value network often remains uncorrelated with the iterative updates of the policy network $\pi_\phi$. As a result, value estimation can be independently executed prior to policy extraction. As delineated in Algorithm 1, to maintain the stability of training for the meta-scoring network, we structured the training process into two distinct phases: the initial phase focuses on value network evaluation following the TD learning of (27), and the subsequent phase engages in bi-level optimization for achieving value-aligned behavior cloning. Although integrating value network learning into the bi-level optimization is feasible, further details on this integration are provided in the Appendix J. Additionally, we incorporated controlled, progressively decreasing noise in the outer loop to facilitate limited exploration during the early stages of training.

To implement our VACO framework, we alternate between the internal loop and external loop optimization. The parameter $\phi$ is only involved in the internal loop and can be easily updated with typical gradient descending approaches, where the $\eta_1$ denotes the learning rate of policy model:

$$\phi_t \leftarrow \phi_{t-1} - \eta_1 \nabla_\phi J_{BC}^w(\phi) \tag{7}$$

The major difficulty of the VACO optimization stems from the external loop to learn parameter $\alpha$ for the meta-scoring neural network. As witnessed in Eq.6, $\alpha$ is coupled into $\phi$ and its gradient can be derived by applying the chain rule and assuming $\frac{\partial \phi_{t-1}}{\partial \alpha} \approx 0$[1]:

$$
\begin{aligned}
\nabla_\alpha J_\pi(\phi) &= \frac{\partial J_\pi(\phi)}{\partial \phi_t} \cdot \frac{\partial \phi_t}{\partial \alpha} \\
&= \frac{\partial J_\pi(\phi)}{\partial \phi_t} \cdot \frac{\partial(\phi_{t-1} - \nabla_\phi J_{BC}^w(\phi_{t-1}))}{\partial \alpha} \\
&\approx -\frac{\partial J_\pi(\phi)}{\partial \phi_t} \cdot \frac{\partial^2 J_{BC}^w(\phi_{t-1}))}{\partial \phi_{t-1} \partial \alpha} \\
&= -\frac{\partial J_\pi(\phi)}{\partial \phi_t} \cdot \frac{\partial[\pi_{\phi_{t-1}}(s) - a]^2}{\partial \phi_{t-1}} \cdot \frac{\partial w_\alpha(s, a, Q_\theta(s, a))}{\partial \alpha}
\end{aligned}
\tag{8}
$$

The above equation is an approximate solution for external loop updating. Thus, we get the update rule of $\alpha$,

---

[1] More details can be found in Appendix C

$$\alpha_t \leftarrow \alpha_{t-1} + \eta_2 \frac{\partial J_\pi(\phi)}{\partial \phi_t} \cdot \frac{\partial^2 J_{BC}^w(\phi_{t-1}))}{\partial \phi_{t-1} \partial \alpha} \tag{9}$$

According to the updating rules based on Eq.7 and Eq.9, we alternately optimize two sets of parameters as in Algorithm 1.

---

**Algorithm 1:** Value-aligned Behavior Cloning via Bi-level Optimization (VACO)

---

**Input:** Fixed offline dataset $\mathcal{D}$, value network $Q_\theta$, policy network $\pi_\phi$, meta-scoring network $w_\alpha$, update steps for value phase $K_1$, update steps for bi-level phase $K_2$

1  **// Value Training Phase**
2  **for** *update step* $k = 1...K_1$ **do**
3  $\quad$ Sample a minibatch sample pairs $(s, a, r, a')$ from $\mathcal{D}$
4  $\quad$ Update value $\theta$ according to IQL's(27) TD learning
5  **end**
6  **// Bi-level Optimization Phase**
7  **for** *update step* $k = 1...K_2$ **do**
8  $\quad$ Sample a minibatch sample pairs $(s, a, r, a')$ from $\mathcal{D}$
9  $\quad$ Fix meta-scoring $\alpha$ and update policy $\phi$ according to Eq.7
10 $\quad$ Fix policy $\phi$ and update meta-scoring $\alpha$ according to Eq.9
11 **end**

---

## 5 EXPERIMENTS

### 5.1 SETTING

**D4RL.** We utilize the MuJoCo and AntMaze domain tasks from the D4RL (11) benchmark for evaluation. Mujoco domain includes a variety of continuous locomotion tasks with dense rewards. Within this domain, we perform experiments in three environments: halfcheetah, hopper, and walker2d. For each environment, we investigate four different v2 datasets, each representing a distinct data quality level: medium, medium-replay, medium-expert, and expert. Consequently, the MuJoCo domain provides an excellent platform for assessing the effects of diverse datasets derived from policies at varying proficiency levels.

**Baselines.** We consider baselines including four different domain to provide a thorough comparison. (1) Classic methods: BC(37), TD3(14), and CQL(28) ; (2) Explicit regularization methods: TD3+BC(13), IQL(27), PRDC(40), TD7(12) and A2PO(38); (3) Implicit regularization methods: MOPO(51), PLAS(55), and EDP(22); (4) Return-conditioned methods: DT(5), DS4(8), and DC(23); Other methods: SAC-RND(35).

**Setup.** We implement our VACO framework by combining the official implementation of (13) and (5). Specifically, value network, policy network and meta-scoring network are all 3-layer MLP. All hidden dimensions of the network are set to 256. Relu (1) activation is performed after each hidden layer. As for training, we adopt Adam optimizer (25) with a learning rate of 3e-5 for meta-scoring network and with a learning rate of 3e-4 for value and policy network. All experiments are conducted on single NVIDIA RTX 3090 GPU.

More detailed experimental setting of D4RL dataset, baselines and hyperparameter setting, are all available in Appendix A.

### 5.2 PERFORMANCE ON MUJOCO DOMAIN

Tab.1 presents the performance outcomes for various algorithms, including other baselines and our VACO model, in offline settings using the D4RL-Mujoco datasets. All scores are normalized, with a value of 100 indicating the performance level of an expert policy, as described by (11).

As shown, we observe the following: (1) Our VACO surpasses the various methods across almost all tasks and varying quality datasets. (2) Some methods exhibit performance deficiencies in certain

Table 1: Averaged normalized scores on MuJoCo locomotion tasks for D4RL dataset. We evaluate 10 times averaged over 5 random seeds, ± standard deviation. The dataset names are abbreviated as follows: 'medium' to 'm', 'medium-replay' to 'm-r', 'medium-expert' to 'm-e', 'expert' to 'e'. The first segment of the table contains classic offline methods, the second segment for explicit regularization methods, the third segment for implicit regularization methods and the fourth segment for return-conditioned decision transformer methods. Our model outperforms various offline RL algorithms on almost all tasks. We mark the best results in **bold** and the second best with an underline for easy comparison. * denotes averaged scores without 'expert' dataset.

| Dataset: | Hopper | | | | HalfCheetah | | | | Walker2D | | | | average | average* |
|---|---|---|---|---|---|---|---|---|---|---|---|---|---|---|
| Method | m | m-r | m-e | e | m | m-r | m-e | e | m | m-r | m-e | e | | |
| TD3+BC | 59.3 | 60.9 | 98.0 | 109.6 | 48.3 | 44.6 | 90.7 | 93.4 | 83.7 | 81.8 | 110.1 | 110.0 | 82.53 | 75.27 |
| IQL | 66.2 | 94.7 | 91.5 | 108.8 | 47.4 | 44.2 | 86.7 | 95.0 | 78.3 | 73.8 | 109.6 | 109.4 | 83.78 | 76.93 |
| CQL | 61.9 | 86.3 | 96.9 | 106.5 | 46.9 | 45.3 | 95.0 | 97.3 | 79.5 | 76.8 | 109.1 | 109.3 | 84.20 | 77.52 |
| _explicit regularization manner methods:_ | | | | | | | | | | | | | | |
| PRDC | **100.3** | 100.1 | 109.2 | - | **63.5** | **55.0** | 94.5 | - | 85.2 | 92.0 | 111.2 | - | - | 90.11 |
| TD7 | 76.1 | 91.1 | 108.2 | - | 58.0 | 53.8 | **104.6** | - | **91.1** | 89.7 | 111.8 | - | - | 87.16 |
| A2PO | 80.3 | 101.6 | 107.4 | - | 47.1 | 44.7 | 95.6 | - | 84.9 | 82.8 | 112.1 | - | - | 84.07 |
| _implicit regularization manner methods:_ | | | | | | | | | | | | | | |
| MOPO | 26.5 | 92.5 | 51.7 | 16.2 | 40.2 | 54.0 | 57.9 | 1.4 | 14.0 | 42.7 | 55.0 | 0.1 | 37.68 | 48.28 |
| PLAS | 32.9 | 27.9 | 111.0 | - | 39.3 | 43.9 | 96.6 | - | 44.6 | 30.2 | 89.6 | - | - | 57.33 |
| EDP | 81.9 | 101.0 | 97.4 | - | 52.1 | 49.4 | 95.5 | - | 6.9 | **94.9** | 110.2 | - | - | 85.48 |
| _return-conditioned manner methods:_ | | | | | | | | | | | | | | |
| DT | 67.6 | 82.7 | 107.6 | 106.3 | 42.6 | 36.6 | 86.8 | 92.4 | 74.0 | 66.6 | 108.1 | 107.6 | 81.58 | 74.73 |
| DS4 | 89.5 | 87.7 | 110.5 | 109.3 | 47.3 | 43.8 | 94.8 | 89.1 | 81.4 | 80.3 | 109.6 | 105.7 | 87.42 | 82.77 |
| DC | 92.5 | 94.2 | 110.4 | 110.5 | 43.0 | 41.3 | 93.0 | 87.5 | 79.2 | 76.6 | 109.6 | 107.8 | 87.13 | 82.2 |
| _Ours_ | | | | | | | | | | | | | | |
| VACO | 97.2 | **102.3** | **112.6** | **114.0** | 60.2 | 51.4 | 98.3 | **100.6** | 88.8 | 92.4 | **114.5** | **112.9** | **95.43** | **90.86** |
| | ±4.2 | ±7.1 | ±2.2 | ±4.5 | ±2.4 | ±1.2 | ±7.2 | ±1.5 | ±1.8 | ±4.9 | ±0.3 | ±0.6 | - | - |

tasks or at specific data levels, whereas VACO consistently achieves competitive performance across the board. For example, MOPO(51) fails on each expert datasets and EDP(22) gets a large decrease on Walker2D medium dataset(11).

## 5.3 PERFORMANCE ON ANTMAZE DOMAIN

Tab.2 presents the performance outcomes of various algorithms on the AntMaze tasks, which primarily test the model's trajectory stitching capability. As shown, our VACO algorithm achieved the highest average score and demonstrated competitive performance across tasks of varying difficulty levels. In contrast, TD3+BC(13) and PLAS(55) failed to perform well on the larger-scale tasks, while SAC-RND(35) and EDP(22) showed significant drops in performance compared to our method on the medium-play and large-play datasets, respectively.

Table 2: Averaged normalized scores on AntMaze tasks. We evaluate 100 times averaged over 5 random seeds, ± standard deviation. Our model outperforms various methods on the average score of all tasks. We mark the best results in **bold** and the second best with an underline for easy comparison.

| Task name | BC | TD3+BC | IQL | CQL | PLAS | SAC-RND | EDP | VACO(ours) |
|---|---|---|---|---|---|---|---|---|
| antmaze-umaze | 65.0 | 66.3 | 83.3 | 74.0 | 70.7 | **97.0** | 94.2 | 96.4 ±0.6 |
| antmaze-umaze-diverse | 55.0 | 53.8 | 70.6 | **84.0** | 45.3 | 66.0 | 79.0 | 83.4 ±9.4 |
| antmaze-medium-play | 0.0 | 26.5 | 64.6 | 61.2 | 16.0 | 38.5 | 81.8 | **82.8** ±3.8 |
| antmaze-medium-diverse | 0.0 | 25.9 | 61.7 | 53.7 | 0.7 | 74.7 | **82.3** | 80.8 ±8.7 |
| antmaze-large-play | 0.0 | 0.0 | 42.5 | 15.8 | 0.7 | 43.9 | 42.3 | **63.8** ±14.8 |
| antmaze-large-diverse | 0.0 | 0.0 | 27.6 | 14.9 | 0.3 | 45.7 | **60.6** | 53.2 ±17.4 |
| Average | 20.0 | 28.7 | 58.3 | 50.6 | 22.3 | 60.9 | 73.4 | **76.7** |

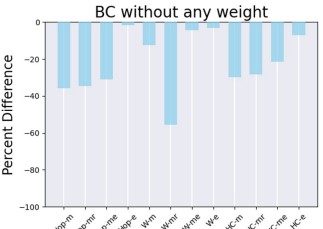 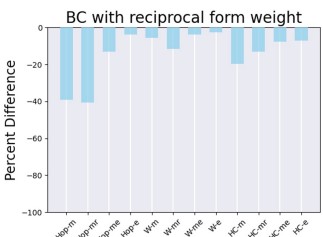 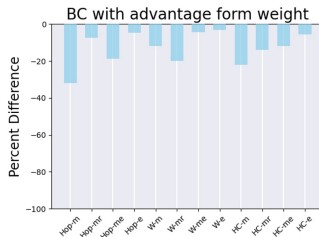

Figure 3: Percent difference of the performance of a comparison among our approach and other heuristic weighting strategy. HC=HalfCheetah, Hop=Hopper, W=Walker, m = medium, mr = medium-replay, me = medium-expert, e = expert. As expected, our model obtains superior results.

## 5.4 COMPARISON WITH OTHER HEURISTIC WEIGHTING STRATEGY

In our work, we introduce a meta-scoring neural network to generate meta weights for weighted behavior learning. One may naturally wonder: can we adopt heuristic weighting strategies, assigning greater weights to samples with larger value estimation, as more efficient alternative? To answer above question, we compare VACO with two heuristic weighting strategies: 1) using the reciprocal of value as weight (13) and 2) using advantage-weight regression (36) as weight. As shown in Fig.3, the two heuristic weighting strategies can boost performance of BC in some situations, but the improvement is far less than VACO on almost all datasets. The reason behind this may be that the heuristic approaches are difficult to achieve alignment between the learned policy and value.

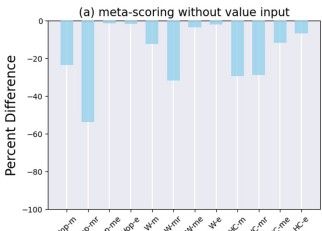 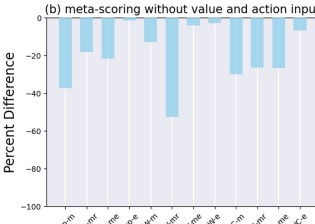 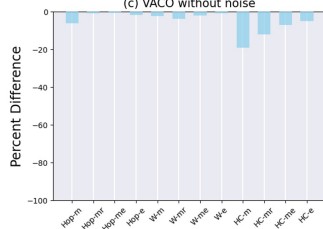

Figure 4: Percent difference of the performance of the ablations of our approach, compared to the full algorithm. HC=HalfCheetah, Hop=Hopper, W=Walker, m = medium, mr = medium-replay, me = medium-expert, e = expert.

## 5.5 ABLATION STUDY ABOUT THE NOISE AND DIFFERENT INPUTS OF META-SCORING NETWORK

To verify the effectiveness of proposed component, we conduct ablation studies on the inputs of meta-scoring network and the controllable noise. In practice, we sequentially remove the value and state from the inputs, and compare the performance with our original VACO configuration. As shown in Fig.4(a) and (b), removing the two parts of input results in a severe performance drop, especially on *medium* and *medium-replay* datasets. This demonstrates the significance of the information within current input for sample weight determination. As shown in Fig.4(c), the gradual reduced noise can boost model performance to some extent, especially in halfcheetah environment, in where limited exploration is encouraged.

## 6 CONCLUSION

We present VACO, a novel bi-level framework to balance OOD problem and value alignment issue concurrently for offline reinforcement learning. The framework comprises of two workflow loop: the internal loop for weighted behavior cloning and the external loop for policy-value alignment. In detail, VACO additionally introduces a meta-scoring network for assigning different importance weights to in-sample offline data. Extensive experiments demonstrate that our proposed VACO achieves

state-of-the-art performance on a variety of continuous control tasks on the D4RL benchmark and the related ablation studies also verify the effectiveness of our method.

ACKNOWLEDGMENTS

This work was supported by the National Natural Science Foundation of China under Grant 62325101 and Grant 62031001, National Key Laboratory of Unmanned Aerial Vehicle Technology in NPU (Grant No.WR202403), and the Fundamental Research Funds for the Central Universities.

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

# A  ADDITIONAL EXPERIMENTAL SETTINGS

## A.1  DETAILS OF D4RL

In our study, we utilize two types of environments in D4RL[2]: Gym Mujoco and Antmaze to conduct our experiments.

### A.1.1  GYM MUJOCO

We focus on three primary environments in Gym Mujoco: Hopper, Half-Cheetah, and Walker2d, as shown in Fig 5. These environments are well-suited for testing the robustness and efficacy of reinforcement learning algorithms across a spectrum of data quality levels, each derived from different policies. We adopt the **v2** datasets ,each reflecting four distinct types: "medium", "medium-replay", "expert", and "medium-expert", reflecting varying degrees of policy optimization and performance.

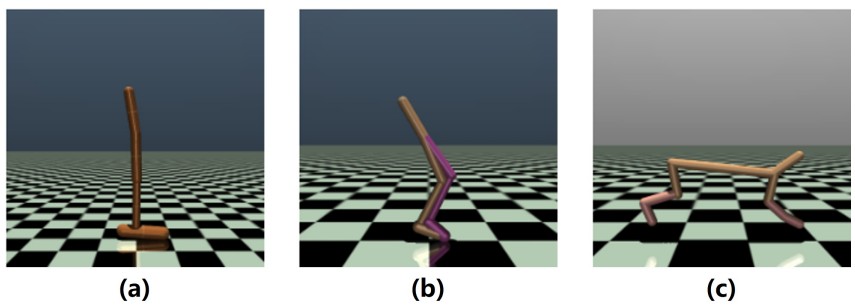

Figure 5: Visulization of (a) Hopper, (b) Walker2d and (c) Half-Cheetah.

**Medium.** Each "medium" dataset is generated by training a policy using the Soft Actor-Critic algorithm. The training process is halted prematurely—specifically, once the policy reaches approximately one-third the performance of an expert policy. This early stopping ensures that the policy is only partially trained, simulating a suboptimal policy performance.

**Medium-replay.** The "medium-replay" datasets consist of all samples stored in the replay buffer during the training phase of the medium policies. This dataset captures a wide array of experiences, both successful and suboptimal, encountered by the policy as it progressed towards its medium-level performance cap.

**Expert.** The "expert" datasets are comprised solely of data derived from a fully trained policy, where the performance is considered to be optimal or near-optimal. These samples provide a benchmark of high-performance behavior in each environment.

**Medium-expert.** The "medium-expert" datasets are constructed by mixing equal parts of samples from expert demonstrations with those from the medium datasets. This mixture offers a unique blend of high-quality and average-quality samples, useful for assessing the algorithm's capability to generalize across different levels of policy proficiency.

### A.1.2  ANTMAZE

The AntMaze domain is a navigation environment that replaces the simple 2D ball from Maze2D with a more complex 8-DoF "Ant" quadruped robot. Data is collected by training a goal-reaching policy, combined with the same high-level waypoint generator from Maze2D. This generator provides subgoals that guide the Ant robot towards the final goal. The terms "umaze," "medium," and "large" refer to different map sizes, as shown in Fig 6.

Specifically, 1) in the "antmaze-umaze" setting, the Ant is tasked with reaching a specific goal from a fixed start location. 2)In the "diverse" datasets, the Ant is directed to a random goal from a random start location. 3) In the "play" datasets, the Ant is guided to specific, hand-picked locations within the

---

[2]D4RL dataset is available at https://github.com/berkeley-rll/d4rl

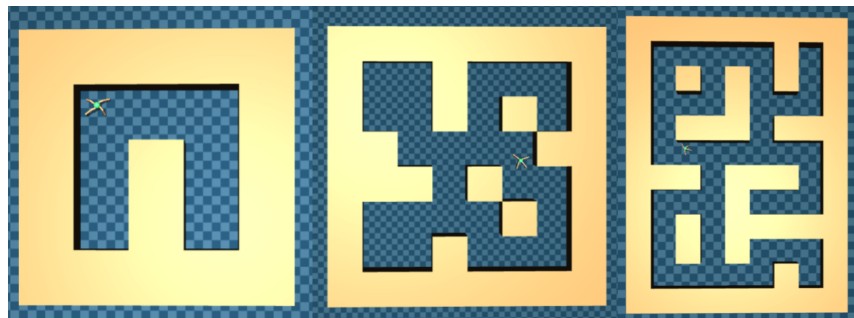

Figure 6: Visualization of the Antmaze environment, with "umaze," "medium," and "large" from left to right.

maze (which are not necessarily the evaluation goal), starting from a different set of hand-picked start locations.

### A.1.3 FRANKA KITCHEN

The Franka Kitchen (16) domain involves controlling a 9-DoF Franka robot in a kitchen environment, as shown in Fig 7, with common household items like a microwave, kettle, overhead light, cabinets, and an oven. The goal of each task is to interact with these items to achieve a desired goal configuration. The "complete" dataset consists of the robot successfully performing all tasks in sequence, making it straightforward for imitation learning methods. In contrast, the "partial" and "mixed" datasets contain undirected data, where the robot performs subtasks unrelated to the goal configuration. In the "partial" dataset, a portion of the data guarantees successful task completion, allowing imitation learning agents to learn from the right subsets selectively. The "mixed" dataset contains no full task completions, requiring RL agents to piece together relevant sub-trajectories to solve the tasks.

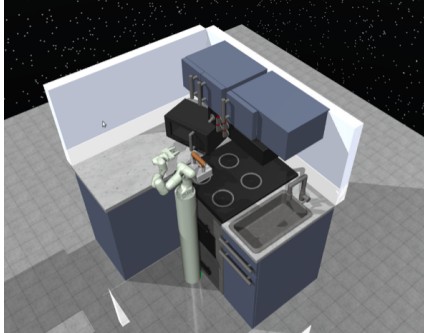

Figure 7: Visualization of the Franka Kitchen environment

### A.1.4 ADROIT

The Adroit domain (39), as shown in Fig 8, involves controlling a 24-DoF Shadow Hand robot to perform four tasks: hammering a nail, opening a door, twirling a pen, and picking up a ball. In the D4RL benchmark, the Adroit domain includes three types of data: human (25 trajectories), expert (large RL policy data), and cloned (data from an imitation policy mixed with demonstrations). The human dataset is designed to test the impact of small amounts of human demonstration data, the expert dataset evaluates the performance with large amounts of expert data, and the cloned dataset simulates a scenario where a small amount of additional data is collected by an imitation policy trained on demonstrations. The domain is challenging due to sparse rewards, high-dimensional tasks, and a narrow expert data distribution, making it a benchmark for evaluating offline RL approaches in realistic, manipulation settings.

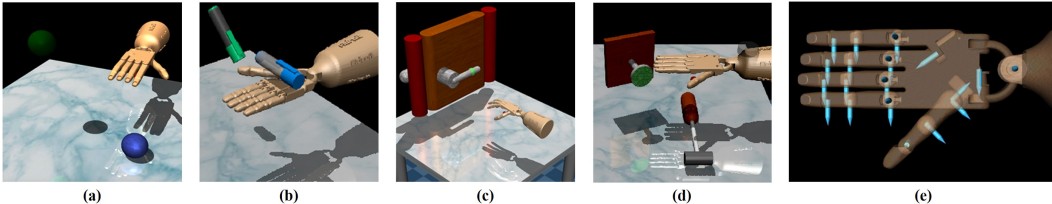

Figure 8: Visualization of the Adroit environment: (a) picking up a ball to a given place; (b) twirling a pen; (c) opening a door; (d) hammering a nail; (e) 24 degree of freedom ADROIT hand.

### A.1.5 V-D4RL

V-D4RL (32) is an offline reinforcement learning dataset based on the D4RL format, designed for vision-based high-dimensional observations. As shown in Fig 9, it includes three tasks from the DMControl Suite: walker-walk, where a planar walker aims to stay upright and maintain a target velocity; cheetah-run, where a biped agent is rewarded for its forward velocity; and humanoid-walk, a challenging task with a 21-jointed humanoid aiming for a target velocity. For each task, we use two levels of data: medium, collected from a medium-performance policy, and expert, collected from a high-performance expert policy.

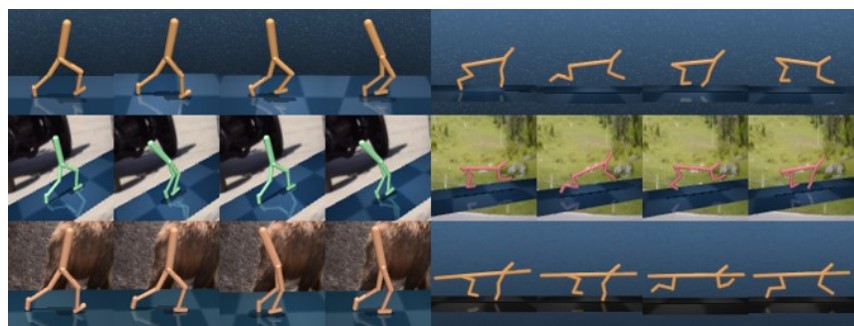

Figure 9: Visualization of the V-D4RL environment.

### A.2 BASELINE DETAILS

We consider baselines including following different domain to provide a thorough comparison. (1) Classic methods: Behavior Cloning (37), TD3 (14), and CQL (28); (2) Explicit regularization methods: TD3+BC (13), IQL (27), PRDC(40), TD7(12) and A2PO(38); (3) Implicit regularization methods: MOPO (51), PLAS (55), and EDP (22); (4) Return-conditioned methods: DT (5), DS4 (8), and DC (23). We obtain baseline performance scores for CQL, TD3+BC and IQL from (27), for DT, DS4 and DC from (23), for BRAC+ from (53), for MOPO from (51), for PLAS from (55), for EDP from (22). For TD3, BC and methods lacking results in *expert* dataset, we reproduce the results using the code provided by the respective authors.

As for the additional Adroit tasks, we add RORL(50) as new baseline. As for Kitchen tasks, we add RvS-G(10) as new baseline.

TD7(12) enhances TD3(14) for offline RL with the State-Action Learned Embeddings (SALE) mechanism, capturing state-action dynamics for better predictions and generalization. It also incorporates a behavior cloning term to maintain policy alignment with the dataset distribution, ensuring stable offline training. SAC-RND(35) extends Soft Actor-Critic (SAC) with Random Network Distillation (RND) to penalize out-of-distribution actions, reducing reliance on ensemble-based methods for efficiency. It uses Feature-wise Linear Modulation (FiLM) to enhance conservatism, ensuring stable offline learning without direct environment interactions. A2PO(38) resolves constraint conflicts in offline RL by using a Conditional Variational Auto-Encoder (CVAE) to disentangle behavior policies based on advantage values. This approach enables prioritization of high-advantage actions while preserving policy diversity, leading to improved advantage estimation and performance on

diverse datasets. PRDC(40) addresses value overestimation in offline RL by regularizing policies to align with the nearest state-action pair in the dataset. This softer point-to-set constraint balances generalization and conservatism, reducing overestimation and improving performance across tasks.

### A.3 IMPLEMENTATION DETAILS OF VACO

We implement VACO by combining the official DT code (https://github.com/kzl/decision-transformer), TD3+BC code (https://github.com/sfujim/TD3_BC) and IQL code (https://github.com/ikostrikov/implicit_q_learning).

## B RUNNING TIME COMPARED WITH OTHER METHODS

Tab.3 presents a comparison of the total runtime of our method against other approaches. We evaluate thest methods using the PyTorch library on a single NVIDIA RTX 3090 GPU for 1e6 update steps. Compared to IQL(27), our method is 1.7 times slower; while compared to DT(5), our method is 2.6 times faster. It is worth mentioning that the meta-scoring network introduced in our framework can be removed during the testing phase, which has no impact on the forward inference time of the policy model.

Table 3: Total running time compared with other methods.

| Methods | BC | TD3+BC | IQL | DT | DC | VACO |
|---|---|---|---|---|---|---|
| total running time | 1.3h | 3.2h | 3.4h | 14.9h | 12.6h | 5.8h |
| scores on locomotion tasks | 64.57 | 82.53 | 83.78 | 81.58 | 87.13 | 95.43 |

## C CLARIFICATION ABOUT $\frac{\partial \phi_{t-1}}{\partial \alpha} \approx 0$

Following recent methods(3; 46; 21) in meta-learning, we make the Markov assumption, under which we assume $\frac{\partial \phi_{t-1}}{\partial \alpha} \approx 0$. This approximation is widely used in the field of bi-level optimization(4; 31) and is referred to as the "one-step differentiation." Under this assumption, at step $t$, given $\phi_{t-1}$ we do not consider about the specific values of $\alpha$ from previous steps that led to $\phi_{t-1}$. Intuitively, this assumption suggests that $\alpha_{t-1}$ has already been updated with respect to $\phi_{t-1}$, so the effect of $\alpha$ on $\phi_{t-1}$ is likely minimal.(46)

Of course, more accurate gradients can be obtained by iteratively applying the update formula for $\phi$ (Eq.7) over multiple steps, known as "K-step differentiation" in bi-level optimization domain. However, it is evident that achieving more precise gradients comes at the cost of significantly increased complexity, both in terms of implementation difficulty and computational/memory overhead. More recent work(4) has demonstrated that the one-step differentiation (i.e., assuming $\frac{\partial \phi_{t-1}}{\partial \alpha} \approx 0$) agrees to an upper bound on the error between the one-step differentiation gradient and the theoretical gradient. For more details, see Corollary 4 in (4). Additionally, in extensive experiments, (4) showed that the one-step differentiation achieves an error of order $10^{-12}$ in logistic regression problem.

Building on the theoretical and practical foundations established by these prior works, we believe that $\frac{\partial \phi_{t-1}}{\partial \alpha} \approx 0$ is valid. This assumption allows for significant simplification and acceleration of computation, within an acceptable range of gradient error. As stated in (46), *"Our use of the Markov assumption is based on its use and empirical success in previous work on bi-level optimization, such as Hyper Gradient Descent(3) and many others. Of course, this is a simplifying assumption, but we believe that our empirical results show that the proposed method is useful nonetheless. Relaxing this assumption would be an interesting avenue for future work. However at the same time how to do so without resulting in large increases in complexity is a challenge that would require additional methodological advantages beyond the scope of the current work."*

# D  ADDITIONAL EXPERIMENTS ON ADROIT AND KITCHEN DOMAIN

We further evaluated the proposed VACO method on the Adroit(39) and Kitchen(16) environments of the D4RL benchmark. Tab.4 and Tab.5 present the experimental results on the Adroit and Kitchen datasets, respectively (partial results adopted from (45), thanks for their awesome work). While our method does not achieve the best performance on certain individual datasets, it consistently demonstrates the best average performance across all datasets compared to other methods, which highlights the competitiveness of our approach.

Table 4: Averaged normalized scores on Adroit tasks. We evaluate 10 times averaged over 3 random seeds, $\pm$ standard deviation. Our model outperforms various methods on the average score of all tasks. We mark the best results in **bold** and the second best with an underline for easy comparison.

| Task Name | BC | BEAR | TD3+BC | IQL | CQL | SAC-RND | RORL | VACO(ours) |
|---|---|---|---|---|---|---|---|---|
| Pen-human | 34.4 | -1.0 | 81.8 ±14.9 | 81.5 ±17.5 | 37.5 | 5.6 ±5.8 | 33.7 ±7.6 | **106.0** ±11.4 |
| Pen-cloned | 56.9 | 26.5 | 61.4 ±19.3 | 77.2 ±17.7 | 39.2 | 2.5 ±6.1 | 35.7 ±3.1 | **98.3** ±15.8 |
| Pen-expert | 85.1 | 105.9 | **146.0** ±7.3 | 133.6 ±16.0 | 107.0 | 45.4 ±22.9 | 130.3 ±4.2 | 144.8 ±4.8 |
| Door-human | 0.5 | -0.3 | -0.1 ±0.0 | 3.1 ±2.0 | **9.9** | 0.0 ±0.1 | 3.78 ±0.7 | 6.1 ±3.8 |
| Door-cloned | -0.1 | -0.1 | 0.1 ±0.6 | 0.8 ±1.0 | 0.4 | 0.2 ±0.8 | -0.1 ±0.1 | **1.6** ±1.5 |
| Door-expert | 34.9 | 103.4 | 84.6 ±44.5 | **105.3** ±2.8 | 101.5 | 73.6 ±26.7 | 104.9 ±0.9 | 104.7 ±1.0 |
| Hammer-human | 1.5 | 0.3 | 0.4 ±0.4 | 2.5 ±1.9 | **4.4** | -0.1 ±0.1 | 2.3 ±1.9 | 1.8 ±0.9 |
| Hammer-cloned | 0.8 | 0.3 | 0.8 ±0.7 | 1.1 ±0.5 | 2.1 | 0.1 ±0.4 | 1.7 ±0.5 | **2.5** ±1.9 |
| Hammer-expert | 125.6 | 127.3 | 117.0 ±30.9 | 129.6 ±0.5 | 86.7 | 24.8 ±39.4 | **132.2** ±0.7 | 127.0 ±0.4 |
| Relocate-human | 0.0 | -0.3 | -0.2 ±0.0 | 0.1 ±0.1 | 0.2 | 0.0 ±0.0 | 0.0 ±0.0 | **0.6** ±0.4 |
| Relocate-cloned | -0.1 | -0.3 | -0.1 ±0.1 | **0.2** ±0.4 | -0.1 | 0.0 ±0.0 | 0.0 ±0.0 | 0.1 ±0.1 |
| Relocate-expert | 101.3 | 98.6 | **107.3** ±1.6 | 106.5 ±2.5 | 95.0 | 3.4 ±4.5 | 47.8 ±13.5 | 107.3 ±3.0 |
| Average | 36.7 | 38.4 | 49.9 | 53.4 | 40.3 | 12.9 | 41.0 | **58.4** |

Table 5: Averaged normalized scores on Kitchen tasks. We mark the best results in **bold** and the second best with an underline for easy comparison.

| Task Name | BC | BCQ | BEAR | CQL | IQL | RvS-G | PLAS | VACO(ours) |
|---|---|---|---|---|---|---|---|---|
| Kitchen-complete | **65.0** | 9.1 | 0.0 | 43.8 | 62.5 | 50.2 | 38.1 | 60.0 |
| Kitchen-partial | 38.0 | 17.6 | 0.0 | 49.8 | 46.3 | **60.3** | 27.0 | 52.5 |
| Kitchen-mixed | 51.5 | 11.5 | 0.0 | 51.0 | 51.0 | 51.4 | 30.0 | **57.5** |
| Average | 51.5 | 12.7 | 0.0 | 48.2 | 53.3 | 54.0 | 31.7 | **56.7** |

# E  ADDITIONAL EXPERIMENTS ON V-D4RL DOMAIN

In addition to the D4RL dataset, we also conducted experiments on the V-D4RL(32) dataset. Similar to D4RL(11), V-D4RL provides analogous problem settings and uses the same data collection process. The key difference is that in V-D4RL, the agent's state observations are replaced with image-based representations of the environment. Following the protocol of the original V-D4RL work(32), we present the corresponding experimental results in Tab.6. Compared to other methods, our proposed approach achieves state-of-the-art or close-to-state-of-the-art performance on most tasks.

Table 6: Averaged normalized scores on V-D4RL tasks. We evaluate 10 times averaged over 3 random seeds, $\pm$ standard deviation. We mark the best results in **bold** and the second best with an underline for easy comparison.

| Environment | Dataset | Offline DV2 | DrQ+BC | CQL | BC | LOMPO | VACO(ours) |
|---|---|---|---|---|---|---|---|
| walker-walk | medium | 34.1 ±19.7 | 46.8 ±2.3 | 14.8 ±16.1 | 40.9 ±3.1 | 43.4 ±11.1 | **53.8** ±1.8 |
| | expert | 4.8 ±0.6 | 68.4 ±7.5 | 89.6 ±6.0 | 91.5 ±3.9 | 5.3 ±7.7 | **92.4** ±3.7 |
| cheetah-run | medium | 17.2 ±3.5 | 53.0 ±3.0 | 40.9 ±5.1 | 51.6 ±1.4 | 16.4 ±8.3 | **53.7** ±0.6 |
| | expert | 10.9 ±3.2 | 34.5 ±8.3 | 61.5 ±4.3 | 67.4 ±6.8 | 14.0 ±3.8 | **75.8** ±6.5 |
| humanoid-walk | medium | 0.2 ±0.1 | 6.2 ±2.4 | 0.1 ±0.0 | 13.5 ±4.1 | 0.1 ±0.0 | **15.3** ±2.2 |
| | expert | 0.2 ±0.1 | 2.7 ±0.9 | 1.6 ±0.5 | **6.1** ±3.7 | 0.1 ±0.0 | 4.3 ±0.8 |
| Average | | | 11.2 | 35.3 | 34.8 | 45.2 | 13.2 | **49.2** |

# F ANALYSIS ABOUT THE RELATIONSHIP BETWEEN META WEIGHT AND Q-VALUE

To examine the relationship between the learned meta-weights and Q-values, we conducted experiments on the Hopper medium-expert dataset, which comprises a combination of expert-level and suboptimal trajectories. As depicted in Fig.14, the meta-weights produced by the meta-scoring network are presented alongside the corresponding Q-value estimates for each state-action pair, visualized through a scatter plot.

For a statistical assessment, we employed the Pearson correlation coefficient to quantify the linear association between Q-values and meta-weights across all state-action pairs in the Hopper medium-expert dataset. The computed correlation coefficient of 0.24 suggests a weak positive correlation, indicating that the meta-scoring network exhibits a tendency to assign higher weights to actions with greater Q-values, while lower weights are generally assigned to actions with smaller Q-values. Nonetheless, the relatively low magnitude of the correlation coefficient reveals that this relationship is not strictly proportional, with some instances deviating from this trend. This also indicates that meta-score is not only affected by Q-value, but also related to state and action.

| type | State-action pairs on Hopper medium-expert dataset | | | | | | | | | |
|---|---|---|---|---|---|---|---|---|---|---|
| Q-value | ⋯ | -1.01 | -0.63 | -0.11 | -0.05 | 0.14 | 0.21 | 0.32 | 0.76 | 1.23 | ⋯ |
| weight | ⋯ | 0.29 | 0.36 | 0.95 | 0.48 | 0.84 | 2.55 | 1.05 | 2.11 | 1.50 | ⋯ |

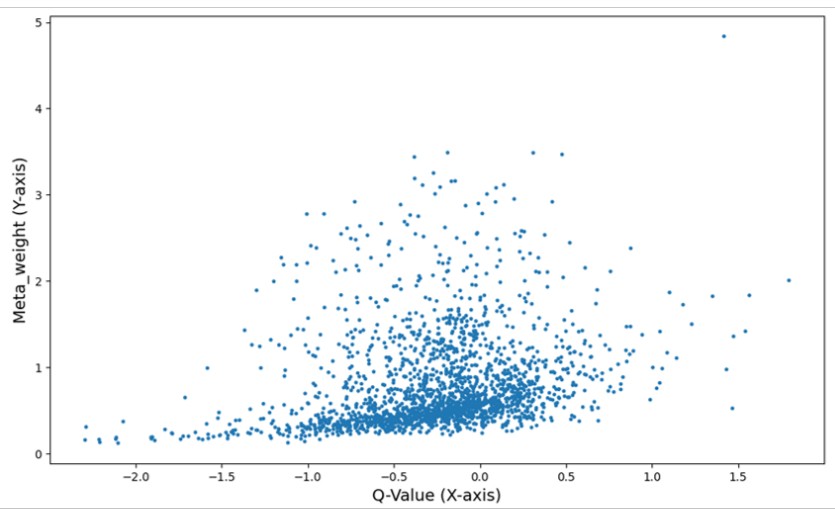

**The Pearson correlation coefficient of weight and Q-value: 0.24**

Figure 10: Visualization of the learned meta weights and corresponding Q-value estimation on Hopper medium-expert dataset. The Pearson correlation coefficient indicates that there is a weak positive correlation between the weights and Q-value.

# G ANALYSIS ABOUT THE RELATIONSHIP BETWEEN META WEIGHT AND TRAJECTORY REWARDS

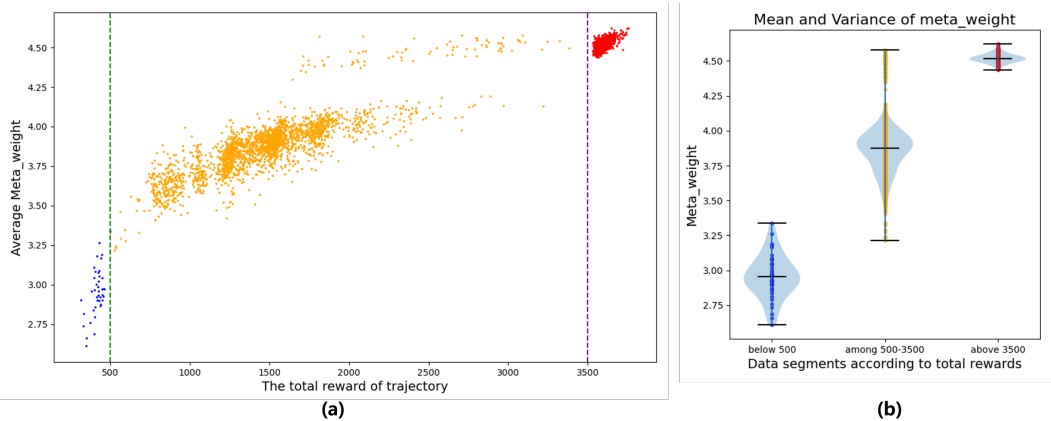

**The Pearson correlation coefficient of meta weight and trajectory total rewards: 0.954**

Figure 11: Visualization of the learned meta weights and corresponding trajectory total rewards on Hopper medium-expert dataset. (a) A scatter plot showing the total trajectory reward versus the average meta-weight of state-action pairs across the entire trajectory. The trajectories are divided into three groups based on the total reward: 1) total reward below 500, 2) total reward between 500 and 3500, and 3) total reward above 3500, representing trajectories generated by "random," "medium," and "expert" policies, respectively. (b) A visualization of the mean and variance distributions of meta-weights corresponding to the three trajectory groups categorized by total reward. The Pearson correlation coefficient indicates that there is a strong positive correlation between the average meta weights and trajectory total rewards.

Appendix F illustrates a weak positive correlation between meta-weights and Q-values, indicating that higher Q-values do not necessarily correspond to higher meta-weights. To further explore the intrinsic nature of the learned meta-weights, we conducted additional experiments on the Hopper medium-expert dataset, focusing on the relationship between the average meta-weight of entire trajectories and their corresponding total rewards.

Specifically, the Hopper medium-expert dataset consists of 3213 trajectories generated by various policies, encompassing a total of 1,999,400 state-action pairs. The total trajectory rewards span the range [315.87, 3759.08]. We categorized the trajectories into three groups based on their total rewards: trajectories with rewards below 500 are considered to originate from "random" policies, those between 500 and 3500 from "medium" policies, and those above 3500 from "expert" policies.

Fig.11(a) presents a scatter plot of the average meta-weight for each trajectory against its total reward. In this visualization, blue points represent trajectories generated by random policies, orange points correspond to medium policies, and red points indicate expert policies. To statistically quantify the linear relationship between average meta-weights and total rewards, we computed the Pearson correlation coefficient, which yielded a value of 0.954. This result demonstrates a strong positive correlation between meta-weights and total rewards.

Fig.11(b) visualizes the mean and variance of the average meta-weights for trajectories across the three policy groups. The results show a clear separation of meta-weights among trajectories generated by different policy groups.

These findings suggest that meta-weights can effectively reflect the quality of the data: higher meta-weights are associated with state-action pairs more likely generated by expert policies, whereas lower meta-weights are linked to those generated by random policies. This is an exciting discovery, as it provides novel insights into the functionality and interpretability of the meta-weights learned by the meta-scoring network. This observation also aligns with the original design intent of the meta-scoring network for distinguishing among various data quality.

# H    ANALYSIS ABOUT THE TRAINING CURVES OF THE INNER AND OUTER LOOP

As shown in Fig.12, we present the training curves of the inner and outer loops on the Hopper expert dataset. During training, the weighted actor loss in the inner loop decreases continuously, while the Q maximization in the outer loop increases until it stabilizes. Notably, when the Q maximization in the outer loop stabilizes, the weighted actor loss in the inner loop still tends to decrease.

This phenomenon aligns with the goal of our designed bi-level optimization framework, where we aim to find an optimal weighted behavioral cloning policy. This policy not only aligns well with the value function (i.e., maximizes Q) but also minimizes the distance to in-sample state-action pairs. Therefore, the relationship between the inner and outer loops is more complementary rather than adversarial.

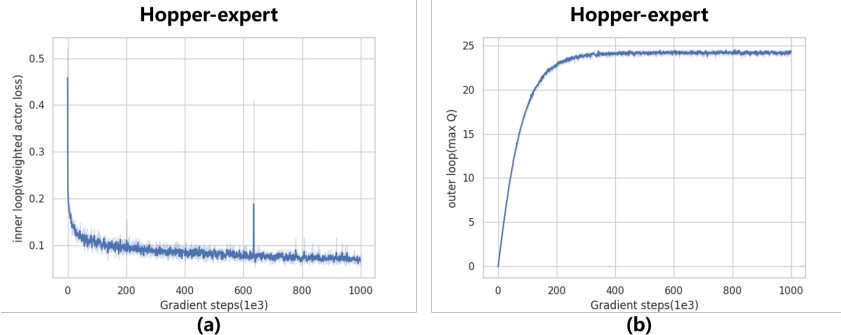

Figure 12: Visualization of the inner loop and outer loop training curves. (a) illustrates the curve of the weighted actor loss function in the inner loop with the number of updates; (b) depicts the curve of Q maximization in the outer loop with the number of updates.

# I  TOY EXAMPLE FOR THE ALIGNMENT OF VACO

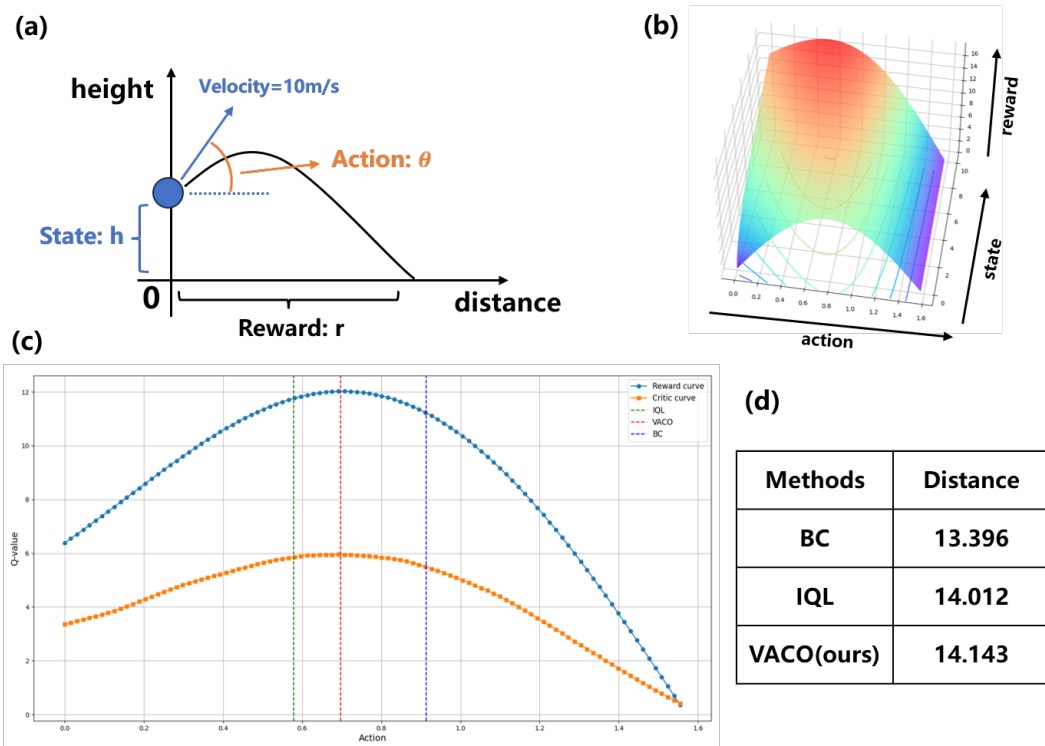

Figure 13: (a) A schematic diagram of the toy example environment; (b) Visualization of the learned Q-value function; (c) Alignment between different methods and the value function when the state (height) is fixed at 2m; (d) Experimental results of various methods in the toy example environment.

To intuitively validate the performance improvements brought by alignment, we designed the toy example environment shown in Fig.13(a). In this environment, a ball is projected with an initial speed of 10 m/s from an initial height $h$ and angle $\theta$. To simplify the modeling, we only consider the effect of gravity. The initial height $h$ serves as the state input, the projection angle $\theta$ as the action, and the final distance traveled by the ball when its height drops to zero as the reward.

For offline reinforcement learning training, we randomly sampled 500 points within the state height range of [0, 10]. For each state, we further randomly sampled 4 points within the action angle range of $[0, \pi/2]$, resulting in a total of 2000 offline samples as the offline dataset. We trained BC, IQL, and VACO on this dataset and tested these three methods at 100 equally spaced points within the state height range of [0, 10]. Fig.13(d) presents the experimental results of BC, IQL, and VACO. Comparatively, our method achieves superior performance.

In the experiments, IQL and VACO share the same value function, and Fig.13(b) visualizes the learned value function. For an intuitive comparison, we plotted the cross-section when the state height is fixed at 2m in Fig.13(c). The horizontal axis represents the action, while the vertical axis represents the value function. The blue curve illustrates the relationship between actions and the true rewards, and the orange curve represents the relationship between actions and the learned value function. The green, red, and blue vertical lines represent the action outputs of IQL, VACO, and BC at a state height of 2m, respectively. It can be observed that the action output by our method is closer to the optimal point of the value curve, resulting in the highest corresponding reward.

## J    MORE DISCUSSION OF VACO

$$\min_{\alpha} \quad J_\pi(\phi)$$

$$\text{s.t. } \phi^*(\alpha), \theta^* = \arg\min_{\phi,\theta} \quad J_{BC}^w(\phi) + J_Q(\theta) \tag{10}$$

We can place the updates of the value network within the internal loop, allowing simultaneous optimization of both the value and policy networks, as demonstrated in Eq.10. As mentioned in the manuscript, since there is no intersection between the value network and either the policy network or the meta-scoring network, introducing value network does not affect the optimization processes of the latter two. The pseudo-code incorporating value updates is presented in Algorithm 2.

As illustrated in Eq.10, VACO can be considered a novel training framework for offline reinforcement learning. In this framework, $J_Q(\theta)$ can be substituted with appropriate value estimation loss function, such as adopted in IQL (27) or CQL (28), and $J_{BC}^w(\phi)$ can be replaced with appropriate behavior cloning-based policy extraction loss function, such as AWR (36). These substitutions indicate the potential applicability of VACO in the field of offline reinforcement learning.

---

**Algorithm 2:** Value-aligned Behavior Cloning via Bi-level Optimization (VACO)

---

**Input:** Fixed offline dataset $\mathcal{D}$, value network $Q_\theta$, policy network $\pi_\phi$, meta-scoring network $w_\alpha$, update steps $K$

1 **for** *update step $k = 1...K$* **do**
2     Sample a minibatch sample pairs $(s, a, r, a')$ from $\mathcal{D}$
3     Update value $\theta$ according to IQL's (27) TD learning
4     Fix meta-scoring $\alpha$ and update policy $\phi$ according to Eq.7
5     Fix policy $\phi$ and update meta-scoring $\alpha$ according to Eq.9
6 **end**

---

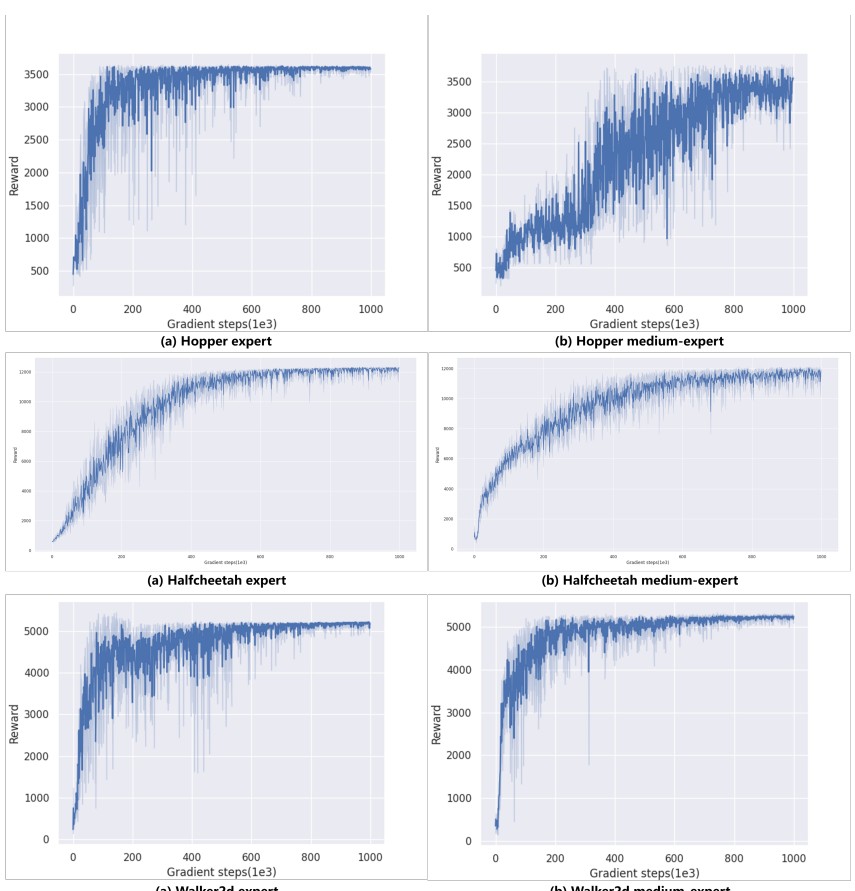

Figure 14: Learning curves for MuJoCo tasks.

| Module | Name | Value |
|---|---|---|
| **Value network** | Batch size | 256 |
| | Learning rate | 3e-4 |
| | Number of layers | 3 |
| | Update steps | 1e6 |
| | Hidden dim | 256 |
| **Policy network** | Batch size | 256 |
| | Learning rate | 3e-4 |
| | Number of layers | 3 |
| | Update steps | 1e6 |
| | Hidden dim | 256 |
| **Meta-scoring network** | Learning rate | 3e-5 |
| | Number of layers | 3 |
| | Hidden dim | 256 |
| **Common** | Discount | 0.99 |
| | Activation function | ReLU |
| | Optimizer | Adam |
| | Adam epsilon | 1e-5 |
| | Weight decay | 1e-6 |
| | $\sigma$ of noise | $1 \rightarrow 0$ |

Table 7: Main hyperparameters of VACO. All experiments were run on a single Nvidia RTX 3090 GPU with Python 3.7 and Torch 1.12.0. Refer to our sample code for full implementation details.

