# OpenReview forum: "Value-aligned Behavior Cloning for Offline Reinforcement Learning via Bi-level Optimization"
_ICLR.cc/2025/Conference — ICLR 2025 Poster_

### Official Review · Reviewer_vQjh · 2024-10-30

**Soundness:** 3
**Presentation:** 3
**Contribution:** 2
**Rating:** 6
**Confidence:** 4

**Summary:**

The paper proposes to address the issue of balancing the value estimation and behaviour cloning in face of data of a mixed quality. The proposed idea is to turn the policy learning and behaviour regression losses to a bi-level optimization. The core idea is to learn a score network to assign a weight to each datapoint in the inner loop such that the learning signal from low-quality data is downplayed. The score network is tuned in the outer loop with a form of policy learning loss with a regularization of adding Gaussian noise to the predicted action. The inner loop is expanded as a one-step update rule so the gradient can be derived. The approach is tested on Hopper, HalfCheetah, Walker2D and AntMaze environments, reporting improvements over a range of baselines with different behavior cloning regularizations.

**Strengths:**

* The idea is well motivated with the challenges of aligning trained behaviour and the one used for value estimation well elaborated.

* The core idea of enforcing a weighted objectives through bi-level optimization is intuitive and easy to understand.

* The selected baselines seem relevant to the comparison and assessment.

**Weaknesses:**

* Insufficient evaluation on more complicated task environments. The selected tasks look simple and could be expanded to cover tasks beyond locomotion. Test results on tasks relying on visual input and manipulation tasks such as Fetch could improve the soundness of the results.

* The results only contain ultimate performance on a limited set of tasks. There is no direct evidence or analysis that how the proposed learning scheme mitigate the alignment issue. A study that can explicitly attributes the improvement to a better alignment could lead to a stronger argument.

* The performance improvement seems diminished on more complicated AntMaze task. This raises the question whether the method can scale up to more complicated scenarios as suggested above.

* Part of the writing on the inner loop derivative is a bit unclear. See the questions below.

**Questions:**

* Can the method be demonstrated to work well on more comprehensive benchmarks including more diverse skills and sensory input?

* Can the observed improvement be linked to a better alignment that was identified as the root cause of inferior performance of previous approaches?

* Can the score network be used to evaluate data snippets to show they are indeed reflecting the optimality of the data?

* Eq. (6) indicates a noise perturbation to the state. Typo?

* Eq. (8) avoids second-order derivatives and it seems due to the assumed $\frac{\partial \phi_{t-1}}{\partial \alpha} \approx 0$. Why this can be valid and what is defined as the Markov process here?

---

> ### Author Response · Authors · 2024-11-21
>
> Dear Reviewer vQjh,
>
> We sincerely appreciate your detailed feedback and constructive suggestions! Here is our response to your concerns:
>
> ---
>
> **Q1: Can the method be demonstrated to work well on more comprehensive benchmarks including more diverse skills and sensory input?**
>
> **A1:** Thank you for your constructive suggestions! We have conduct additional experiments on more complicated adroit and kitchen task environments beyond locomotion in D4RL, as detailed in the revised manuscript's **Appendix D, Table 4 and Table 5**. Moreover, we also conduct experiments on the V-D4RL dataset, which shares similar problem settings as D4RL, but with image-based representations of the environment as input, as detailed in the revised manuscript's **Appendix E, Table 6**. While our method may not achieve the best performance on every individual dataset, it demonstrates superior average performance across tasks, which shows the competitiveness of VACO. Due to limited time and resources, we are still making our best effort to extend our method to manipulation tasks. Once the relevant experiments are completed, we will include the results in the revised manuscript.
>
> ---
>
> **Q2: Can the observed improvement be linked to a better alignment that was identified as the root cause of inferior performance of previous approaches?**
>
> **A2:** Thank you for your constructive suggestions! We have conduct relevant experiments on a toy environment to intuitively verify whether a better alignment is the reason of better performance, as detailed in the revised manuscript's **Appendix I, Figure 13**. It can be observed that the action output by our method is closer to the optimal point of the value function, resulting in the highest corresponding reward.
>
> ---
>
> **Q3: Can the score network be used to evaluate data snippets to show they are indeed reflecting the optimality of the data?**
>
> **A3:** Good question!
>
> From the perspective of network design, the goal of the score network is to assign weights to each in-sample data point to find the best weighted behavior cloning policy, achieving alignment with the value network. This implies that the score network is likely to assign higher weights to higher-value data points within the in-sample domain.
>
> From an experimental standpoint, we have conducted relevant analyses on the relationship between meta-weights and Q-values, as detailed in **Appendix F, Figure 10** of the revised manuscript. For a statistical analysis, we employed the Pearson correlation coefficient to quantify the association between Q-values and meta-weights. The computed coefficient is 0.24, indicating a weak positive correlation between meta-scores and Q-values. Moreover, as suggested by Reviewer BNkz, we have conducted relevant analyses on the relationship between meta-weights and cumulative trajectory rewards across different policies, as detailed in the revised manuscript's **Appendix G, Figure 11**. The computed Pearson correlation coefficient yielded a value of 0.954, indicating a strong positive correlation. This finding suggests that the learned meta-weights effectively capture the quality of the data: higher meta-weights are associated with state-action pairs more likely generated by "expert" (high-quality) policies, while lower meta-weights correspond to those generated by "random" (low-quality) policies.
>
> Based on above analysis, we believe the score network can reflect the optimality of data to some extent.
>
> ---
>
> **Q4: Eq. (6) indicates a noise perturbation to the state. Typo?**
>
> **A4:** This is not a typo. We intentionally add Gaussian noise, which gradually decays as the number of update steps increases, to the input state of the policy network $\pi_{\phi}$. This is designed to enable limited exploration in the action space.

---

> > ### Author Response · Authors · 2024-11-21
> >
> > ---
> >
> > **Q5: Eq. (8) avoids second-order derivatives and it seems due to the assumed $\frac{\partial \phi_{t-1}}{\partial \alpha}\approx 0$. Why this can be valid and what is defined as the Markov process here?**
> >
> > **A5:** Thank you for your constructive suggestions!Here, we would like to clarify two points.
> >
> > Firstly, the avoidance of second-order derivative computation in Eq. (8) is not due to the assumption that $\frac{\partial \phi_{t-1}}{\partial \alpha} \approx 0$. In fact, the primary source of second-order derivative computation arises from the term $\frac{\partial^2 J_{BC}^{w}(\phi_{t-1})}{\partial \phi_{t-1} \partial \alpha}$ in Eq. (8). From the definition of $J_{BC}^{w}(\phi_{t-1})$ in Eq. (5), it is evident that since the meta-weights and behavior cloning loss are related multiplicatively, the computation of the second-order derivative can be reduced to a matrix multiplication of two first-order derivatives.
> >
> > Secondly, following recent methods [1,2,3] in meta-learning, we make the Markov assumption, under which we assume $ \frac{\partial \phi_{t-1}}{\partial \alpha} \approx 0 $. This approximation is widely used in the field of bi-level optimization [4,5] and is referred to as the "one-step differentiation." Under this assumption, we treat the iterative updates of $ \frac{\partial \phi}{\partial \alpha} $ as a Markov process, meaning that at step t, given $\phi_{t-1}$, we do not consider the specific values of $ \alpha $ from previous steps that led to $ \phi_{t-1} $.
> >
> > The main purpose of this approximation is to simplify the computation of $\frac{\partial \phi_{t}}{\partial \alpha}$, thereby reducing the optimization complexity. Theoretically, recent work [4] has demonstrated that one-step differentiation (i.e., assuming $\frac{\partial \phi_{t-1}}{\partial \alpha} \approx 0 $) provides an upper bound on the error between the gradient obtained from one-step differentiation and the theoretical gradient of $ \frac{\partial \phi_{t}}{\partial \alpha} $. For more details, see Corollary 4 in [4]. Furthermore, in experiments, [4] demonstrated that the one-step differentiation achieves an error of order $10^{-12}$ in logistic regression problems.
> >
> > Building on the theoretical and practical foundations established by these prior works, we believe that $\frac{\partial \phi_{t-1}}{\partial \alpha} \approx 0 $ is valid. As stated in [2], *"Our use of the Markov assumption is based on its use and empirical success in previous work on bi-level optimization, such as Hyper Gradient Descent [1] and many others. Of course, this is a simplifying assumption, but we believe that our empirical results show that the proposed method is useful nonetheless."*
> >
> > We have synchronously included the above clarifications in the revised manuscript's **Appendix C**.
> >
> >
> > [1] AG Baydin et al., "Online learning rate adaptation with hypergradient descent." ICLR'2018
> >
> > [2] X Wang et al., "Optimizing data usage via differentiable rewards." ICML'2020
> >
> > [3] X Jiang et al., "Boosting supervised dehazing methods via bi-level patch reweighting." ECCV'2022
> >
> > [4] J Bolte et al., "One-step differentiation of iterative algorithms." NIPS'2024
> >
> > [5] R Liu et al., "A generic first-order algorithmic framework for bi-level programming beyond lower-level singleton." ICML'2020

---

> > > ### Comment · Reviewer_vQjh · 2024-11-22
> > >
> > > I would like to thank the authors' diligence and efforts to address the comments. I think the new results improved the paper.
> > >
> > > Specifically,
> > > 1. The validation on more diverse tasks, though not decisively favours the proposed method, provides more support on the its generality and potential.
> > > 2. The new study on the correlation and the discussion with Reviewer BNkz provides direct evidence to justify the introduction of the proposed score-network.
> > > 3. The Markovian assumption for one-step differentiation is elaborated. Although I still have some doubts on its damage to the theoretical rigour, I am fine to take it as a simplification for implementation in practice.
> > >
> > > I am now leaning towards the positive side and update the rate.

---

### Official Review · Reviewer_M9oj · 2024-10-30

**Soundness:** 4
**Presentation:** 3
**Contribution:** 4
**Rating:** 8
**Confidence:** 5

**Summary:**

This work proposes a novel algorithm, VACO, that combines two offline RL approaches: behavior cloning and value estimation, in a meta-learning framework. In particular, the outer loop uses value estimation to train the weights of behavioral cloning data samples. The approach is designed to balance common issues in offline reinforcement learning: namely, the over-estimation of the value of out-of-distribution state-action pairs, and convergence on sub-optimal actions in-distribution.

**Strengths:**

The motivation is clear: the issues with current offline RL methods are clearly established.

The approach is original: it combines two well-established methods in a novel way that produces better results than either method alone.

SOTA performance on an established benchmark.

**Weaknesses:**

The related works section (Section 5) is placed after the experiments (Section 4), despite describing the methods compared in the experiments. This made the experiments section much harder to parse through on the first pass. Placing this information somewhere before Section 3 could greatly improve the paper’s clarity on the first readthrough.

The writing is hard to parse in a few places. In the abstract (lines 25-26), the description of inner and outer loops appears to suggest that the outer loop only adds noise, rather than training the weighting, which it does according to the description in 3.3.

There is no analysis on training costs (number of iterations required or wall time). Given the use of meta-learning, I would assume that the approach is generally more costly than most of the alternatives. Because of the competitive performance of the algorithm, it doesn’t need to be faster than other methods, but cost would be an important concern in determining when to use this approach versus a simpler one.

**Questions:**

For the experiments, is the same value network used across all approaches in each environment? Or does each approach have a separately trained value network?

---

> ### Author Response · Authors · 2024-11-20
>
> Dear Reviewer M9oj,
>
> We sincerely appreciate your positive comments and constructive suggestions! In response to your questions, we have prepared the answers below:
>
> ---
>
> **W1: The related works section (Section 5) is placed after the experiments (Section 4), despite describing the methods compared in the experiments. This made the experiments section much harder to parse through on the first pass. Placing this information somewhere before Section 3 could greatly improve the paper’s clarity on the first readthrough.**
>
> **A1:** We appreciate the reviewer’s suggestion regarding the structure of the paper. We agree that placing the related works section before Section 3 could indeed improve the clarity of the paper on the first read. Specifically, we will place it after the introduction and before Section 3 in the revised manuscript, ensuring smoother readability and better context for understanding the experiments.
>
> ---
>
> **W2: The writing is hard to parse in a few places. In the abstract (lines 25-26), the description of inner and outer loops appears to suggest that the outer loop only adds noise, rather than training the weighting, which it does according to the description in 3.3.**
>
> **A2:** Thank you for pointing out the ambiguity in the abstract. The outer loop not only introduces controlled noise for limited exploration but also serves to align the policy with the value function by maximizing the value estimation. We have rephrased the abstract to clarify that the outer loop's primary function is value alignment, and the noise addition is a secondary mechanism for exploration during training. The revised sentence is: “the outer loop maximizes value estimation for alignment with controlled noise to facilitate limited exploration.” The specific modifications can be found in the blue text section of the revised manuscript's **abstract**.
>
> ---
>
> **W3: There is no analysis on training costs (number of iterations required or wall time). Given the use of meta-learning, I would assume that the approach is generally more costly than most of the alternatives. Because of the competitive performance of the algorithm, it doesn’t need to be faster than other methods, but cost would be an important concern in determining when to use this approach versus a simpler one.**
>
> **A3:** We agree that a discussion on training costs is crucial for determining the practical applicability of our approach. As detailed in the revised manuscript's **Appendix B, Table 3**, We conduct the experiments of total running time compared with other methods with PyTorch Library on single NVIDIA RTX 3090 GPU. Table 3 shows that while VACO is 1.7 times slower than IQL, it is 2.6 times faster than DT.
>
> ---
>
> **Q1: For the experiments, is the same value network used across all approaches in each environment? Or does each approach have a separately trained value network?**
>
> **A4:** In practice, each approach has a separately trained value network.

---

> > ### Comment · Reviewer_M9oj · 2024-11-25
> >
> > Thank you for your response and for taking my suggestions into consideration. I have updated the scores of the individual sections accordingly. I will keep the score a solid 8, but I will increase my confidence score.

---

### Official Review · Reviewer_8RhL · 2024-10-31

**Soundness:** 3
**Presentation:** 3
**Contribution:** 4
**Rating:** 8
**Confidence:** 4

**Summary:**

Summary:

This paper introduces a novel bi-level offline DRL framework called value-aligned behavior Closing via bi-level optimization (VACO).
The authors identify critical challenges in offline RL, particularly out-of-distribution (OOD) and value alignment issues.
To address these issues, the VACO framework employs a bi-level optimization strategy integrating a meta-scoring neural network to generate adaptive weights for behavior learning.
In the various tasks in the D4RL benchmark, VACO achieves state-of-the-art performance compared to other SOTA methods.

**Strengths:**

Strengths

1. This paper's writing is excellent. It utilizes illustrative figures to clearly demonstrate the out-of-distribution (OOD) and value alignment issues in offline DRL and effectively explains the motivation behind the method. The methodology and experimental setup are clearly presented, with an appendix and code provided for additional details.

2. This approach combines the maximization of the value estimation function with weighted behavioral cloning from a novel perspective, proposing a bi-level learning framework that simultaneously mitigates OOD issues and value alignment issues.

3. The method was tested across multiple tasks on the D4RL benchmark. The results are impressive, as the method, using a simple MLP, surpasses most other methods, including diffusion networks, and achieves the highest average performance.

**Weaknesses:**

Weaknesses

1. The name of the method may not be entirely appropriate. Although it includes "bi-level optimization," in lines 9 and 10 of the algorithm, the two networks are alternately trained without forming an outer and inner loop. This approach resembles adversarial training, similar to the generator and discriminator in GANs. A more suitable name might be "Adversarial/Complementary Value-Aligned Behavior Cloning."

2. Could the authors further explore the relationship between maximizing the value estimation function and weighted behavioral cloning in Equation 6? For instance, do they exhibit an adversarial relationship, or are they complementary? It would be helpful to provide training loss curves or other visualizations of the two losses to clarify this relationship.

3. Could the authors provide additional results for tasks in the adroit and kitchen environments in D4RL, as well as learning curves for MuJoCo tasks?

4. I am very curious about the relationship between the meta-score and Q-value for a given state-action pair. Could you provide some statistical analysis? Do they exhibit a proportional relationship, such that a larger Q-value corresponds to a higher meta-score? If so, could the Q-value be used as a substitute for the meta-score?

**Questions:**

Please check the weakness section.

---

> ### Author Response · Authors · 2024-11-20
>
> Dear Reviewer 8RhL,
>
> We sincerely appreciate your positive comments and constructive suggestions! In response to your questions, we have prepared the answers below:
>
> ---
>
> **W2: Could the authors further explore the relationship between maximizing the value estimation function and weighted behavioral cloning in Equation 6? For instance, do they exhibit an adversarial relationship, or are they complementary? It would be helpful to provide training loss curves or other visualizations of the two losses to clarify this relationship.**
>
> **A1:** Thanks for your insightful suggestion! We have conduct relevant experiments to further explore the relationship between maximizing the value estimation function and weighted behavioral cloning, as detailed in the revised manuscript's **Appendix H, Figure 12**. As shown in Figure 12, we present the training curves of the inner and outer loops on the Hopper expert dataset. During training, the weighted actor loss in the inner loop decreases continuously, while the Q maximization in the outer loop increases until it stabilizes. This aligns with the goal of our designed bi-level optimization framework, where we aim to find an optimal weighted behavioral cloning policy, which not only aligns well with the value function (i.e., maximizes Q) but also minimizes the distance to weighted in-sample state-action pairs. Therefore, the relationship between the inner and outer loops is more complementary rather than adversarial.
>
> ---
>
> **W1: The name of the method may not be entirely appropriate. Although it includes "bi-level optimization," in lines 9 and 10 of the algorithm, the two networks are alternately trained without forming an outer and inner loop. This approach resembles adversarial training, similar to the generator and discriminator in GANs. A more suitable name might be "Adversarial/Complementary Value-Aligned Behavior Cloning."**
>
> **A2:** As discussed in **A1** above,  the relationship between the inner and outer loops is more complementary rather than adversarial. We are very glad to change our name to "Complementary Value-Aligned Behavior Cloning".
>
> ---
>
> **W3: Could the authors provide additional results for tasks in the adroit and kitchen environments in D4RL, as well as learning curves for MuJoCo tasks?**
>
> **A3:** We have conduct additional experiments for different tasks in the adroit and kitchen environments in D4RL, as detailed in the revised manuscript's **Appendix D, Table 4 and Table 5**. Moreover, we also conduct experiments on the V-D4RL dataset, which shares similar problem settings as D4RL, but with image-based representations of the environment as input, as detailed in the revised manuscript's **Appendix E, Table 6**. While our method may not achieve the best performance on every individual dataset, it demonstrates superior average performance across tasks, which shows the competitiveness of VACO. As for learning curves for MuJoCo tasks, please see **Figure 14** in the revised manuscript.
>
> ---
>
> **W4: I am very curious about the relationship between the meta-score and Q-value for a given state-action pair. Could you provide some statistical analysis? Do they exhibit a proportional relationship, such that a larger Q-value corresponds to a higher meta-score? If so, could the Q-value be used as a substitute for the meta-score?**
>
> **A4:** This is a constructive problem! We have conduct relevant experiments about relationship between meta-weight and Q-value as detailed in the revised manuscript's **Appendix F, Figure 10**. For a statistical analysis, we adopt the Pearson correlation coefficient to quantify the association between Q-values and meta-weights on the Hopper medium-expert dataset. The computed correlation coefficient is 0.24, which suggests a weak positive correlation of meta-score and Q-value but not strictly proportional. Additionally, we also conduct experiments about using Q-value as a substitute for the meta-score and the results are shown below:
>
> |    dataset  |  Hopper medium     |   Hopper medium-replay   | Hopper medium-expert | Hopper expert |
> |---|---|---|---|---|
> | VACO| 97.2 | 102.3| 112.6|114.0 |
> |substitute Q-value for meta-score| 53.6|34.7 |59.2 |111.5 |
>
> As shown above, meta-score obtains better performance than Q-value.
>
> Moreover, as suggested by Reviewer BNkz, we have conducted relevant analyses on the relationship between meta-weights and cumulative trajectory rewards across different policies, as detailed in the revised manuscript's **Appendix G, Figure 11**. The computed Pearson correlation coefficient yielded a value of 0.954, indicating a strong positive correlation. This finding suggests that the learned meta-weights effectively capture the quality of the data: higher meta-weights are associated with state-action pairs more likely generated by "expert" (good) policies, while lower meta-weights correspond to those generated by "random" (bad) policies, providing new insights into the interpretability of the meta-weights.

---

### Official Review · Reviewer_BNkz · 2024-11-03

**Soundness:** 3
**Presentation:** 3
**Contribution:** 2
**Rating:** 6
**Confidence:** 4

**Summary:**

In offline RL, behavior constraints (BC objectives) and value estimation (Q-values) are commonly used strategies to train policies. However, when both strategies are applied together, they tend to interfere with each other, often resulting in sub-optimal performance. To address this issue, the paper proposes VACO, which first learns the value network (Q-function) and then employs a bi-level learning process: training a meta-score network in the outer loop, and updating the policy using a weighted BC objective in the inner loop. The proposed VACO method is compared with several baselines on tasks from the D4RL dataset. Experimental results demonstrate that VACO achieves superior performance across tasks and various data resource settings.

**Strengths:**

- **Paper Writing:** The paper has a coherent and concise narrative. The challenge is clearly described, the proposed method is easy to understand, and the figures and tables are well-designed, allowing readers to quickly grasp the key points.

- **Method Originality:** Although applying learnable models/parameters to dynamically adjust the weight of each training sample is not a new idea, the insight to develop bi-level optimization in the context of behavior constraints in offline RL is underexplored, lending sufficient originality to the method.

- **Reproducibility:** The inclusion of source code and detailed experimental information enhances the paper’s reproducibility.

**Weaknesses:**

**Major Concerns**
- **Theroem Correctness:** While the proposed method seems to have enough originality, I have doubts about its supporting theorem, particularly the assumption on line 261, i.e. $\frac{\partial\phi_{t-1}}{\partial\alpha} \approx 0$. The paper indicates that this assumption is supported by the Markov property. However, if I understand correctly, the Markov property primarily concerns state transitions in a stochastic process, where the future state depends only on the current state (and action, in the case of an MDP) and not on past states. In this scenario, however, it involves dependencies between functions across iterations, not probabilistic state transitions. I believe the divergence of $\phi_t$ and $\phi_{t-1}$ should be evaluated to better determine whether this assumption is valid.
- **Evaluation Tasks:** The proposed method and baselines are only examined on simple tasks from one benchmark, i.e., D4RL, making it unclear whether the comparison results hold sufficient generalization. I recommend conducting the evaluation on Adroit tasks (also from D4RL) at least, or on more complex benchmarks such as Robomimic [R1], which also provides collected trajectories with different levels of expertise.
- **Ablation Studies:** The paper lacks in-depth ablations. While conducting the study on removing inputs of the meta-score network is a nice start, it is more basic. More comprehensive ablations, such as visualizing the relationship between assigned weights and the state-action pairs collected by different policies, could better demonstrate the effectiveness of the proposed method.
- **Baseline Details:** The paper lacks descriptions of the selected baselines. Although the paper categorizes baselines into four groups, there is no explanation of each baseline's key insights or features, raising concerns about whether the compared baselines are robust enough.

---

**Minor Concerns**
- **Task Setting:** While I acknowledge that offline RL is a critical area, especially considering safety and training efficiency, the setting studied in the paper is somewhat dated; the policy can only learn from a single collected dataset. Recent offline RL works have explored more diverse directions, such as auxiliary unlabeled data [R2, R3], primitive discovery [R4], or offline-to-online settings [R5], which may have the potential to create a greater impact.

- **Experimental Results:** As mentioned in the appendix, some results from previous works are copied. I suspect this is also the reason for some missing fields in Table 1. I suggest conducting experiments for all methods under the same conditions (same devices and random seeds) to better demonstrate the fairness of the comparison.

- **Typos:** While the paper is generally well-written, there are some typos that should be corrected. For instance:

  line 094, line 478: "In details" -> "In detail"

  line 227: "Gauss noise" -> "Gaussian noise"

  line 236: "square error" -> "squared error"

  line 239: "Through such bi-level optimization framework" -> "Through such a bi-level optimization framework"

  line 307: (1) classical methods: (1) Explicit regularization methods: -> This line needs clarification or correction.

Note: Addressing these concerns is highly encouraged, but they did not substantially affect my evaluation.

---

[R1] Ajay Mandlekar et al., "What Matters in Learning from Offline Human Demonstrations for Robot Manipulation," CoRL'21.

[R2] Konrad Zolna et al., "Offline Learning from Demonstrations and Unlabeled Experience," NeurIPS'20.

[R3] Tianhe Yu et al., "How to Leverage Unlabeled Data in Offline Reinforcement Learning," ICML'22.

[R4] Anurag Ajay et al., "OPAL: Offline Primitive Discovery for Accelerating Offline Reinforcement Learning," ICLR'21.

[R5] Shenzhi Wang et al., "Train Once, Get a Family: State-Adaptive Balances for Offline-to-Online Reinforcement Learning," NeurIPS'23.

**Questions:**

1. Could you please clarify how the assumption on line 261 is derived? I believe the Markov property may not be an appropriate basis for this assumption.
2. Does the proposed VACO demonstrate sufficient generalization ability? Can it still achieve superior performance on more challenging tasks, such as those in the Adroit or Robomimic datasets?
3. Can the learned meta-score network assign low weights to state-action pairs collected by policies that exhibit poor performance? Any visualizations or discussions would be helpful.
4. Please provide more details about the compared baselines so we can better assess the superiority of the proposed method.
5. From line 748 to line 753, the paper states that any value estimation loss function and any behavior cloning-based policy extraction loss function can be used. Is this claim supported by any experiments?
6. Why is Gaussian noise used in Eq. 6? Additionally, why is the noise injected into the state rather than into the intermediate latent variables or the output logits?

---

> ### Comment · Reviewer_BNkz · 2024-11-18
>
> Dear Authors of Submission 11212,
>
> After reviewing the comments from other reviewers, I noticed several shared concerns, particularly regarding the **theoretical support** and the **limited evaluation tasks**.
>
> To me, theoretical correctness is more important than the method's practical  performance. Therefore, I strongly recommend that the authors address my raised concerns, especially questions 1, 2, 3, and 5.
>
> If these questions are satisfactorily addressed, I would be happy to raise my score. However, if certain critical concerns remain unresolved, it may influence my final evaluation.
>
> I will remain active during the discussion and look forward to having a constructive conversation with you.
>
> Best,
>
> Reviewer BNkz

---

> > ### Author Response · Authors · 2024-11-20
> >
> > Dear Reviewer BNkz,
> >
> > We sincerely appreciate your active feedback and constructive suggestions! Here is our response to your concerns:
> >
> > ---
> >
> > **Q1: Could you please clarify how the assumption on line 261 is derived? I believe the Markov property may not be an appropriate basis for this assumption.**
> >
> > **A1:** Thanks for your constructive suggestion! Following recent methods[1,2,3] in meta-learning, we make the Markov assumption, under which we assume $\frac{\partial \phi_{t-1}}{\partial \alpha}\approx 0$. This approximation is widely used in the field of bi-level optimization[4,5] and is referred to as the "one-step differentiation." Under this assumption, at step t, given  $\phi_{t-1}$  we do not consider about the specific values of $\alpha$  from previous steps that led to $\phi_{t-1}$. Intuitively, this assumption suggests that $\alpha_{t-1}$ has already been updated with respect to $\phi_{t-1}$, so the effect of  $\alpha$ on $\phi_{t-1}$ is likely minimal.
> >
> > Of course, more accurate gradients can be obtained by iteratively applying the update formula for $\phi$ (Eq.7 in the manuscript) over multiple steps, known as "K-step differentiation" in bi-level optimization domain. However, it is evident that achieving more precise gradients comes at the cost of significantly increased complexity, both in terms of implementation difficulty and computational/memory overhead. More recent work[4] has demonstrated that the one-step differentiation (i.e., assuming $\frac{\partial \phi_{t-1}}{\partial \alpha}\approx 0$) agrees to an upper bound on the error between the one-step differentiation gradient and the theoretical gradient of $\frac{\partial \phi_{t}}{\partial \alpha}$. For more details, see Corollary 4 in [4]. Additionally, in experiments, [4] showed that the one-step differentiation achieves an error of order $10^{-12}$ in logistic regression problem.
> >
> > Building on the theoretical and practical foundations established by these prior works, we believe that $\frac{\partial \phi_{t-1}}{\partial \alpha}\approx 0$ is valid. This assumption allows for significant simplification and acceleration of computation, within an acceptable range of gradient error. As stated in [2], *"Our use of the Markov assumption is based on its use and empirical success in previous work on bi-level optimization, such as Hyper Gradient Descent[1] and many others. Of course, this is a simplifying assumption, but we believe that our empirical results show that the proposed method is useful nonetheless."* Relaxing this assumption would be an interesting avenue for future work. However at the same time how to do so without resulting in large increases in complexity is a challenge that would require additional methodological advantages beyond the scope of the current work.
> >
> > We have synchronously included the above clarifications in the revised manuscript's **Appendix C** and line 261.
> >
> > [1] AG Baydin et al., "Online learning rate adaptation with hypergradient descent." ICLR'2018
> >
> > [2] X Wang et al., "Optimizing data usage via differentiable rewards." ICML'2020
> >
> > [3] X Jiang et al., "Boosting supervised dehazing methods via bi-level patch reweighting." ECCV'2022
> >
> > [4] J Bolte et al., "One-step differentiation of iterative algorithms." NIPS'2024
> >
> > [5] R Liu et al., "A generic first-order algorithmic framework for bi-level programming beyond lower-level singleton." ICML'2020
> >
> > ---
> >
> > **Q2: Does the proposed VACO demonstrate sufficient generalization ability? Can it still achieve superior performance on more challenging tasks, such as those in the Adroit or Robomimic datasets?**
> >
> > **A2:** We have conduct additional experiments for different tasks in the adroit and kitchen environments in D4RL, as detailed in the revised manuscript's **Appendix D, Table 4 and Table 5**. Moreover, we also conduct experiments on the V-D4RL dataset, which shares similar problem settings as D4RL, but with image-based representations of the environment as input, as detailed in the revised manuscript's **Appendix E, Table 6**. While our method may not achieve the best performance on every individual dataset, it demonstrates superior average performance across tasks, which shows the competitiveness of VACO.

---

> > > ### Author Response · Authors · 2024-11-20
> > >
> > > ---
> > >
> > > **Q3: Can the learned meta-score network assign low weights to state-action pairs collected by policies that exhibit poor performance? Any visualizations or discussions would be helpful.**
> > >
> > > **A3:** This is a constructive problem! We have conduct relevant experiments about relationship between meta-weight and Q-value as detailed in the revised manuscript's **Appendix F, Figure 10**. For a statistical analysis, we adopt the Pearson correlation coefficient to quantify the association between Q-values and meta-weights on the Hopper medium-expert dataset. The computed correlation coefficient is 0.24, which suggests a weak positive correlation of meta-score and Q-value, indicating that the meta-scoring network exhibits a tendency to assign higher weights to actions with greater Q-values, while lower weights are assigned to actions with smaller Q-values. Nonetheless, the relatively low magnitude of the correlation coefficient reveals that this relationship is not strictly proportional, with some instances deviating from this trend. We also conduct an additional experiment about using Q-value as a substitute for the meta-score, please see refer to A4 to Reviewer 8RhL.
> > >
> > > ---
> > >
> > > **Q4: Please provide more details about the compared baselines so we can better assess the superiority of the proposed method.**
> > >
> > > **A4:** We guess that the reviewer may have overlooked the **Related Works section 5**, as it was placed later in the old version manuscript. In the **Related Work section 5**, we provided a brief introduction to each baseline. Additionally, in the revised manuscript's **Appendix A.2**, we have included more detailed descriptions of the compared methods.
> > >
> > > ---
> > >
> > > **Q5: From line 748 to line 753, the paper states that any value estimation loss function and any behavior cloning-based policy extraction loss function can be used. Is this claim supported by any experiments?**
> > >
> > > **A5:** Thank you for your constructive suggestions! We try to convey from line 748 to line 753 in the old version manuscript is that VACO could be better recognized as a flexible bi-level framework that can work with appropriate offline RL approaches for balancing the OOD issues and misalignment problems. In this work, we select the IQL's value objective function to apply our bi-level framework. As a supplement, we conduct additional experiments with CQL's value objective function and the results are shown below:
> > >
> > > |dataset|Hopper medium|Hopper medium-replay|Hopper medium-expert|Hopper expert|
> > > |---|---|---|---|---|
> > > |CQL|61.9|86.3|96.9|106.5|
> > > |CQL+VACO|82.3|88.2|106.8|110.4|
> > >
> > > As shown, VACO can improve the performance of CQL to some extents.
> > >
> > > ---
> > >
> > > **Q6: Why is Gaussian noise used in Eq. 6? Additionally, why is the noise injected into the state rather than into the intermediate latent variables or the output logits?**
> > >
> > > **A6:** From an empirical perspective, we chose the widely used Gaussian noise. The noise added during training gradually decreases to zero, similar to the epsilon-greedy strategy. At the early stages of training, the noise facilitates exploration, while its influence is reduced in the later stages to stabilize learning. Adding noise to the intermediate latent variables or the output logits is also a promising direction worth exploring, and we will definitely consider it in future work.
> > >
> > > ---
> > >
> > > **Q7: While I acknowledge that offline RL is a critical area, especially considering safety and training efficiency, the setting studied in the paper is somewhat dated; the policy can only learn from a single collected dataset. Recent offline RL works have explored more diverse directions, such as auxiliary unlabeled data [R2, R3], primitive discovery [R4], or offline-to-online settings [R5], which may have the potential to create a greater impact.**
> > >
> > > **A7:**  Thank you for your constructive suggestions! Exploring the application of the bi-level optimization framework in directions such as auxiliary unlabeled data [R2, R3], primitive discovery [R4], or offline-to-online settings [R5] is highly appealing, and we will definitely consider these extensions in future work.
> > >
> > > ---
> > >
> > > **Q8: As mentioned in the appendix, some results from previous works are copied. I suspect this is also the reason for some missing fields in Table 1. I suggest conducting experiments for all methods under the same conditions (same devices and random seeds) to better demonstrate the fairness of the comparison.**
> > >
> > > **A8:** Thank you for your constructive suggestions! Due to time and resource constraints, we are making our best effort to reproduce the results of the missing methods under the same conditions. Once completed, we will update the table with the results.
> > >
> > > ---
> > >
> > > **typos:** Thank you for pointing out the typos. We have corrected all noted issues in the revised manuscript.

---

> > > > ### Comment · Reviewer_BNkz · 2024-11-21
> > > > **Response to Authors' Rebuttal**
> > > >
> > > > I appreciate the authors' efforts to address my concerns, I have the following comments.
> > > >
> > > > **Q1:** Thank you for providing these additional references. I have checked the original submission, and in line 261, there were no referenced papers. Therefore, I initially thought it referred to the Markov property of MDPs in RL. I will review these references over the next few days to determine if the assumption makes sense to me. However, intuitively, I still feel it may not be appropriate to make such an assumption. As you quoted from [2], it is a "simplified assumption," and even if it may not cause serious issues in logistic regression problems, that does not necessarily mean it will work well in the context studied in this work.
> > > >
> > > > **Q3:** Thank you for the additional experimental results. However, I think it would be more interesting to see the correlation between the weights and the policy used to collect the transitions. As you mentioned, a good action does not guarantee a high Q-value, and I think that is why the correlation coefficient is somewhat low.
> > > >
> > > > In addition to the above two points, my other concerns have been well addressed. I truly appreciate the effort. I will hold my score for now and decide whether it should be updated at the end of the discussion.

---

> > > > > ### Author Response · Authors · 2024-11-21
> > > > >
> > > > > Dear Reviewer BNkz,
> > > > >
> > > > > We sincerely appreciate your active feedback!
> > > > >
> > > > > As for **Q1:**  We sincerely thank the reviewer for his/her time and effort. We have incorporated the newly added references into the revised manuscript to enhance the soundness of our method. Admittedly, the Markov assumption is a simplified hypothesis. However, regardless of this assumption, the approximation $\frac{\partial \phi_{t-1}}{\partial \alpha} \approx 0$ is widely employed in one-step differentiation methods within the field of bi-level optimization, enabling the optimization of the hyperparameter network. If you encounter any issues or have questions while reviewing the provided references, please feel free to discuss with us.
> > > > >
> > > > > As for **Q3:** Thank you for your constructive suggestions! We are slightly puzzled by the expression "the policy used to collect the transitions." Could you clarify whether this refers to the policy used to generate the offline reinforcement learning dataset or the policy learned through training? From our perspective, determining the meta-weight used to evaluate the quality of state-action pairs is inherently complicated and depends on multiple factors, such as state, action, and Q-value. This complexity is precisely why we chose to employ a hyperparameter network within a bi-level optimization framework to adaptively learn these weights.
> > > > >
> > > > > We greatly appreciate your time and effort in reviewing our work and look forward to your further response.  We hope that, through our joint efforts, we can further improve our manuscript to a better version.

---

> > > > > > ### Comment · Reviewer_BNkz · 2024-11-21
> > > > > >
> > > > > > Dear Authors,
> > > > > >
> > > > > > Thank you for your timely comments and inquiry.
> > > > > >
> > > > > > Regarding **Q3**, I am referring to the first case (i.e., the behavior policy used to collect the offline dataset). Since you are adding a behavior constraint/regularization to the learning policy, it is intuitive that we would prefer assigning greater weight to transitions collected by an expert policy rather than a random policy. I would like to explore whether the meta-score network can effectively capture this distinction. Due to the nature of the reward function, these transitions may not always have high Q-values, as evidenced by your new experiment.
> > > > > >
> > > > > > A practical study could involve using the meta-score network, after training, to generate weights for transitions and comparing the average or distribution of these weights for transitions collected by an expert policy versus a random policy. (Each weight could be visualized with a color corresponding to the policy that collected the transition.)
> > > > > >
> > > > > > Of course, an expert policy may occasionally generate suboptimal actions. However, in general, such a comparison should provide better insight into whether the meta-score network can distinguish "worse actions." This capability aligns with the purpose of introducing the weighted BC constraint, right?
> > > > > >
> > > > > > If anything remains unclear, please feel free to let me know.

---

> > > > > > > ### Author Response · Authors · 2024-11-22
> > > > > > >
> > > > > > > Dear Reviewer BNkz,
> > > > > > >
> > > > > > > Thank you for your detailed explanation and insightful suggestions! We fully agree with the idea of exploring the relationship between meta-weights and the policy used to collect the transitions. We have conducted relevant experiments, as detailed in the revised manuscript's **Appendix G, Figure 11**, and have reached exciting findings that we are eager to share with you.
> > > > > > >
> > > > > > > We continued our experiments on the Hopper medium-expert dataset, which contains 3213 trajectories of varying quality, with cumulative rewards ranging from 315.87 to 3759.08. These trajectories were divided into three groups: those with cumulative rewards below 500 were classified as generated by "random" policies, those between 500 and 3500 as generated by "medium" policies, and those above 3500 as generated by "expert" policies. To statistically quantify the relationship between meta-weights and cumulative rewards across different policies, we computed the Pearson correlation coefficient, which yielded a value of **0.954**. This result demonstrates a strong positive correlation between meta-weights and cumulative rewards across different policies. The visualization of these results can be found in **Appendix G, Figure 11**.
> > > > > > >
> > > > > > > These findings indicate that the learned meta-weights effectively reflect the quality of the data: higher meta-weights are associated with state-action pairs more likely generated by "expert"(good) policies, while lower meta-weights correspond to those generated by "random"
> > > > > > > (bad) policies. This is an exciting discovery as it provides novel insights into the interpretability of the meta-weights learned by the meta-scoring network. Furthermore, this aligns indeed with the original design intent of the meta-scoring network—to distinguish among data of varying quality.
> > > > > > >
> > > > > > > We sincerely appreciate your unique perspective and highly valuable feedback，which have greatly contributed to enhancing the depth and completeness of our work! This discovery is truly remarkable, and we will highlight it in the revised version of the manuscript.
> > > > > > >
> > > > > > > Once again, we thank you for your time, thoughtful insights and constructive suggestions. Should you have any further questions or additional feedback, please feel free to let us know. We look forward to continuing this constructive exchange.
> > > > > > >
> > > > > > > Best,
> > > > > > >
> > > > > > > Authors of Submission 11212

---

> > > > > > > > ### Comment · Reviewer_BNkz · 2024-11-22
> > > > > > > >
> > > > > > > > Dear Authors,
> > > > > > > >
> > > > > > > > Thank you for sharing the results with me! They align perfectly with what I expect when the meta-score network functions effectively. I believe this serves as more intuitive evidence for the proposed method's empirical effectiveness.
> > > > > > > >
> > > > > > > > Additionally, I have quickly reviewed the newly provided references (apologies for not being able to review them thoroughly—my schedule is quite packed this week). Interestingly, reference [2] was also questioned on this assumption during its review process. However, since I have not carefully reviewed all the references, I will refrain from challenging the established assumption in the field of bi-level optimization.
> > > > > > > >
> > > > > > > > Considering the new evidence and the effort you have put into the rebuttal, I am happy to raise my score to 6. Thank you again for the timely and constructive discussions.
> > > > > > > >
> > > > > > > > Great job, and good luck!

---

### Meta-Review · Area_Chair_V2KV · 2024-12-15

**Metareview:**

This paper addresses the challenge of balancing behavior cloning (BC) and value estimation in offline reinforcement learning, particularly when dealing with mixed-quality data. It proposes a bi-level optimization framework, VACO, which employs a meta-scoring network in the outer loop to assign adaptive weights to data samples used for BC in the inner loop, ensuring better value alignment. Experimental results on D4RL benchmarks demonstrate that VACO consistently outperforms several baseline methods across diverse tasks and data quality settings.

The strengths of this paper include its clear writing and motivation, the originality of applying bi-level optimization, strong experimental performance, and the inclusion of code and experimental details for reproducibility. The main weaknesses of this paper include the limited evaluation scope on more complex tasks and insufficient theoretical analysis. However, the authors have addressed many of these issues during rebuttal and discussion.

The four reviews are all positive (6, 6, 8, and 8), suggesting that this paper is a clear acceptance.

**Additional Comments On Reviewer Discussion:**

Reviewer BNkz raised concerns about the theoretical analysis, which the authors addressed during the rebuttal, leading to an increased score.
Reviewer vQjh requested more evaluation results, which the authors provided, prompting the reviewer to raise their score.

---

### Decision · Program_Chairs · 2025-01-22

Accept (Poster)